# Efficient candidate drug target discovery through proteogenomics in a Scottish cohort

Jurgis Kuliesius[1], Paul R. H. J. Timmers [1,2], Pau Navarro [2,3], Lucija Klaric [2] & James F. Wilson [1,2] ✉

Understanding the genomic basis of human proteomic variability provides powerful tools to probe potential causal relationships of proteins and disease risk, and thus to prioritise candidate drug targets. Here, we investigated 6432 plasma proteins (1533 previously unstudied in large-scale proteomic GWAS) using the SomaLogic (v4.1) aptamer-based technology in a Scottish population from the Viking Genes study. A total of 505 significant independent protein quantitative trait loci (pQTL) were found for 455 proteins in blood plasma: 382 *cis*- (P < 5×10⁻⁸) and 123 *trans*- (P < 6.6×10⁻¹²). Of these, 31 *cis*-pQTL were for proteins with no previous GWAS. We leveraged these pQTL to perform causal inference using bidirectional Mendelian randomisation and colocalisation against complex traits of biomedical importance. We discovered 43 colocalising associations (with a posterior probability > 80% that pQTL and complex traits share a causal variant), pointing to plausible causal roles for the proteins. These findings include hitherto undiscovered causal links of leukocyte receptor tyrosine kinase (LTK) to type-2 diabetes and beta-1,3-glucuronyltransferase (B3GAT1) to prostate cancer. These new connections will help guide the search for new or repurposed therapies. Our findings provide strong support for continuing to increase the number of proteins studied using GWAS.

Genome-wide association studies (GWAS) are set to significantly impact the rapidly evolving domain of personalised medicine. This specialised area is dedicated to recognising the genetic and other variation among individuals, guiding the way towards precise risk evaluations and subsequent therapies tailored to distinct genetic (and other) profiles[1].

The field of proteomics adds another dimension to our understanding. The proteome participates in virtually every biological process, playing a critical role in both health and disease, with proteins serving as structural components, enzymes, signalling molecules, and more. Elucidating the genetic determinants of protein abundance is essential for understanding the complex interplay of genes, proteins, and their downstream effects on human physiology and disease susceptibility. This paves the way for the development of new prognostic markers[2], drug repurposing[3], and Precision Medicine[4] approaches.

Recent advances in technology have dramatically increased the number of proteins that can be quantified to over 10,000[5], with single proteomics studies exceeding sample sizes of 50,000[6]. The competing aptamer, antibody and mass spectrometry technologies differ in their mode of action,

throughput, and the number of protein targets. Enabled by these innovations, GWAS, Mendelian Randomisation (MR) and comparison of local genetic architectures (colocalisation) are employed to unravel the complex relationships between circulating plasma protein levels and phenotypes such as disease risk. Leveraging naturally occurring genetic variants as instruments allows the assessment of the effects of lifelong exposure to altered protein levels on disease susceptibility, in a conceptually comparable way to performing a randomised control trial.

In this study, we present GWAS of 6432 proteins, representing one of the most comprehensive protein-centric association analyses to date[7]. We then use the resulting 31 *cis* associations from a little-studied 1533-protein subset and explore connections with medically relevant traits and diseases. Post-GWAS analyses resulted in 43 promising links between protein abundance and phenotype, 7 of which we highlight due to their potential therapeutic relevance for future in-depth follow-up. Our primary aim is to identify novel genetic loci associated with protein abundance, thereby uncovering new regulatory mechanisms, and shedding light on the interplay between genetic variants and disease, mediated by the human proteome.

[1]Centre for Global Health Research, Usher Institute, University of Edinburgh, 5-7 Little France Road, Edinburgh, UK. [2]MRC Human Genetics Unit, University of Edinburgh, Western General Hospital, Edinburgh, UK. [3]The Roslin Institute, University of Edinburgh, Easter Bush Campus, Midlothian, UK. ✉e-mail: jim.wilson@ed.ac.uk

## Results

### Discovery of pQTLs

We performed genome-wide association analysis of over 5.4 million imputed common autosomal single nucleotide polymorphisms (SNPs) in 200 individuals using 7596 aptamers targeting 6432 blood plasma proteins measured with the SomaLogic v4.1 assay (Fig. 1). Two different genome-wide significance thresholds were used: $p < 5 \times 10^{-8}$ for *cis* associations, defined as being within 1 Mb from the gene encoding the targeted protein) and $p < 6.58 \times 10^{-12}$ for *trans* associations, defined as all non-*cis* associations. After pruning SNPs with low allele frequency (MAF < 0.05, due to low sample size), we identified a total of 1478 significant associations for the levels of 455 proteins. This corresponded to 505 independent "sentinel" SNPs, as determined by clumping (Fig. 2). A total of 76% (382/505) of the sentinel SNPs were *cis* associations (Supplementary Data 1). In total, 333 proteins had only *cis* associations, 117 only *trans*, with 5 proteins having at least 1 *cis* and *trans* signal. The level of genomic inflation was well controlled for all 7596 aptamers, with the median $\lambda$ value of 1.005, standard error 0.015.

The majority, 82% (412/505), of the independent, sentinel SNPs were associated with a single protein. Six genomic regions were associated with 5 or more protein measurements (Fig. 3, vertical lines). These regions contained the *CFH*, *HRG*, *BCHE*, *ABO*, *VTN* and *APOE* genes, which have already been discovered as pleiotropic hubs or hotspots in previous proteomics studies[8–10].

Given that at the time of writing there were no other GWAS with the SomaLogic 4.1 assay using European ancestry samples, we replicated our results using the published data from Fenland, a large independent European ancestry cohort that utilised an earlier version of the SomaLogic assay[10]. We focused specifically on replicating *cis* associations due to their direct biological relevance to protein levels and their central role in our downstream analyses. After applying filters to ensure unambiguous matching to protein level measurements (see 'Methods'), 272 out of 338 proteins with genome-wide significant *cis* pQTL identified here were able to be tested for replication. All 272 proteins had significant ($p < = 1.0 \times 10^{-11}$) *cis* pQTL associations in the Fenland study. Additionally, there was also a good consistency of effect size and direction (Pearson $r^2 = 0.96$) between the pQTL reported in the two studies (Supplementary Data 1 and Supplementary Fig. 1). Moreover, while full replication was not expected due to differences in ethnicity and population characteristics (general population vs. disease-cohort; British vs. African American ancestry), we investigated whether our *cis* pQTL were in LD with the sentinel SNPs reported by Surapaneni et al.[8], the only other published GWAS with the SomaLogic 4.1 to date. A total of 156 of our *cis* pQTL were found to be in high LD ($r^2 \geq 0.6$) with sentinel *cis* pQTL reported in their study (Supplementary Data 2). Surapaneni et al. also reported *cis* associations for 212 of the 338 proteins in this study. Of these, 131 likely capture the same association signal, as indicated by high LD ($r^2 \geq 0.6$) between sentinel SNPs. However, for 81 of the proteins with *cis* pQTL, the effect on protein levels is likely driven by different underlying causal variants, as evidenced by low LD between sentinel SNPs ($r^2 < 0.6$) or the sentinel variant being absent from the reference panel.

Our analysis reveals 31 novel cis-pQTL, such as those for B3GAT1 (beta-1,3-glucuronyltransferase 1), DCC (Deleted in Colorectal Cancer netrin 1 receptor) and LTK (leukocyte receptor tyrosine kinase), allowing instrumentation of these proteins in Mendelian randomisation analyses.

Notably, some of the genome-wide significant associations were likely due to a degree of amino-acid sequence homology between the aptamer-targeted protein of interest and its paralogues. The strongest non-hub *trans*-pQTL detected in this study was on chromosome 6 (rs11155297, $p = 5.4 \times 10^{-55}$, Fig. 3 and Supplementary Data 1) was associated with FUCA1 (alpha-L-fucosidase 1) protein levels. However, the pQTL maps within the genomic region of *FUCA2* (alpha-L-fucosidase 2) on chromosome 1, the gene product of which shares 55% amino-acid sequence homology with the measured FUCA1 protein, when analysed with Clustal-O[11]. Meanwhile, there was no suggestive association detected within the *FUCA1 cis* genomic region ($p > 1 \times 10^{-5}$). Hence, we conclude that in these

examples the aptamer might not be able to distinguish between the two paralogous proteins and therefore the observed strong *trans* association might in fact be a *cis* association of a mislabelled protein.

Furthermore, two different aptamers targeting FCGR2B (Immunoglobulin G Fc Gamma receptor IIb) also had significant *cis*-associations in the vicinity of the nearby *FCGR2A* (Fc gamma receptor IIa) and *FCGR2C* (Fc gamma receptor IIc) coding regions, while exhibiting 73% and 89% amino-acid sequence homology with the encoded proteins, respectively (Supplementary Fig. 2). The connection between *FCGR2B* and *FCGR2C* may also be due to linkage disequilibrium (LD), with top SNPs, rs17413015 and rs61801833 (73 kb apart), exhibiting an LD $r^2$ value of 0.56, as evaluated with LDproxy[12]. In contrast, the sentinel SNPs, rs17413015 and rs4657041, in *FCGR2B* and *FCGR2A*, respectively, show a lower LD $r^2$ of 0.06, across the 166 kb between them.

As in other studies[9,13], an inverse relationship between the minor allele frequency and the absolute effect size was observed for both *cis* and *trans* associations. Overall, *trans* associations displayed both smaller effect sizes and were less detectable at lower allele frequencies (Fig. 4A). Moreover, there was a strong influence of the distance from the transcription start site on the effect size of the *cis*-pQTL, with both the number of associations and their effect size rapidly decreasing outside the 0.15 Mb range (Fig. 4B).

We next annotated our sentinel pQTLs with the functional consequence information by considering the most severe consequence of any variant that is in $r^2 > 0.8$ with our sentinel variants. Thirty-two out of 505 variants in this study have a high impact (loss-of-function) on the protein structure (e.g. stop/start gain/lost, frameshift). These were not distinguishable in their protein-level variances explained from the 229 protein-altering variant group of moderate impact (in-frame insertion/deletion, missense), $p = 0.731$. High and moderate impact genetic variants showed a significantly stronger effect on protein levels compared to 235 low-impact variants, with $p$ values of $2.21 \times 10^{-3}$ and $2.98 \times 10^{-8}$, respectively. This effect was observed for both *cis* and *trans* associations (Supplementary Fig. 3).

### Protein–disease links

To assess possible causal connections between plasma protein levels and disease outcomes or risk, we next performed bidirectional two-sample Mendelian Randomisation (MR). We focused on the proteins that have not yet been reported in large-scale proteomic MR studies, by cross-referencing the proteins targeted by the SomaLogic v4.1 assay with those measured with the SomaLogic v4.0 and Olink Explore 1536[14] assays. 1533 of the 6432 proteins quantified with the SomaLogic v4.1 assay were not measured in previous large-scale proteomics studies[10,14–18] (Supplementary Data 3 and 4) and did not have associations uncovered through GWAS that could be used as instrumental variables.

We further restricted our selection of instrumental variables to *cis* sentinel pQTL, which are near the genes encoding their respective proteins. This approach was intended to mitigate the impact of pleiotropy on our findings, as a genetic locus in *cis* is less likely to influence multiple unrelated phenotypic traits, thereby simplifying the interpretation of resulting causal relationships. In addition, the focus on *cis* pQTL effectively reduced the multiple testing burden.

Thirty-one of the 1533 proteins introduced in the SomaLogic v4.1 assay had *cis* associations in this study and were used as exposures in forward MR, while a curated list of 3772 diseases and risk factors (see Methods) from the MRC IEU OpenGWAS database[19] were used as outcomes. We found statistically significant associations (FDR < 0.01) for 17 (out of 31) proteins and 95 outcomes (149 protein-outcome pairs). Next, we assessed the possibility of reverse causality by running reverse MR, with the outcomes as exposures and the proteins as outcomes. There was no evidence of reverse causality (reverse MR $p$ value > 0.01) for any of the 149 significant protein-outcome pairs (Supplementary Data 5).

Given that only a single instrumental variable was used for each of these proteins (their *cis*-pQTL), to further validate our findings we next performed a colocalisation analysis, using the "coloc" R package[20]. Colocalisation compares the local architecture of association for each trait in a

**Recruitment**

200 healthy individuals of isolated ancestry in the Scottish Isles of Shetland.

**Proteomic analysis**

SomaScan v4.1, evaluating 6432 unique protein targets in blood plasma.

**Genomic analysis**

>14 million imputed common autosomal variants.

**GWAS**

505 independent pQTL (76% cis) for 455 proteins.

**Unstudied proteome**

31 cis pQTL for 31 proteins not analysed in previous large-scale proteomics studies.

**Replication subset**

All 357 cis pQTL were genome-wide significant and directionally concordant in a matched large-scale cohort.

**Pleiotropic region characterisation**

117 trans pQTL associations replicated 6 pleiotropic regions observed in SomaScan assays.

**Bidirectional Mendelian Randomization and Colocalisation**

43 causal associations for 14 proteins with medical outcomes sharing the same genetic signal.

**Fig. 1 | Study design flowchart depicting the key analyses performed.** This study profiled 6432 plasma proteins in 200 individuals from an isolated Scottish population using the SomaScan v4.1 assay. A genome-wide association scan identified 505 independent pQTL for 455 proteins. Of these, 357 *cis*-pQTL replicated in a large-scale cohort, 31 mapped to previously unstudied proteins, and 6 known pleiotropic regions were found among 117 *trans* associations. Follow-up Mendelian randomisation and colocalisation analyses linked 14 proteins to 43 disease traits through shared genetic signals.

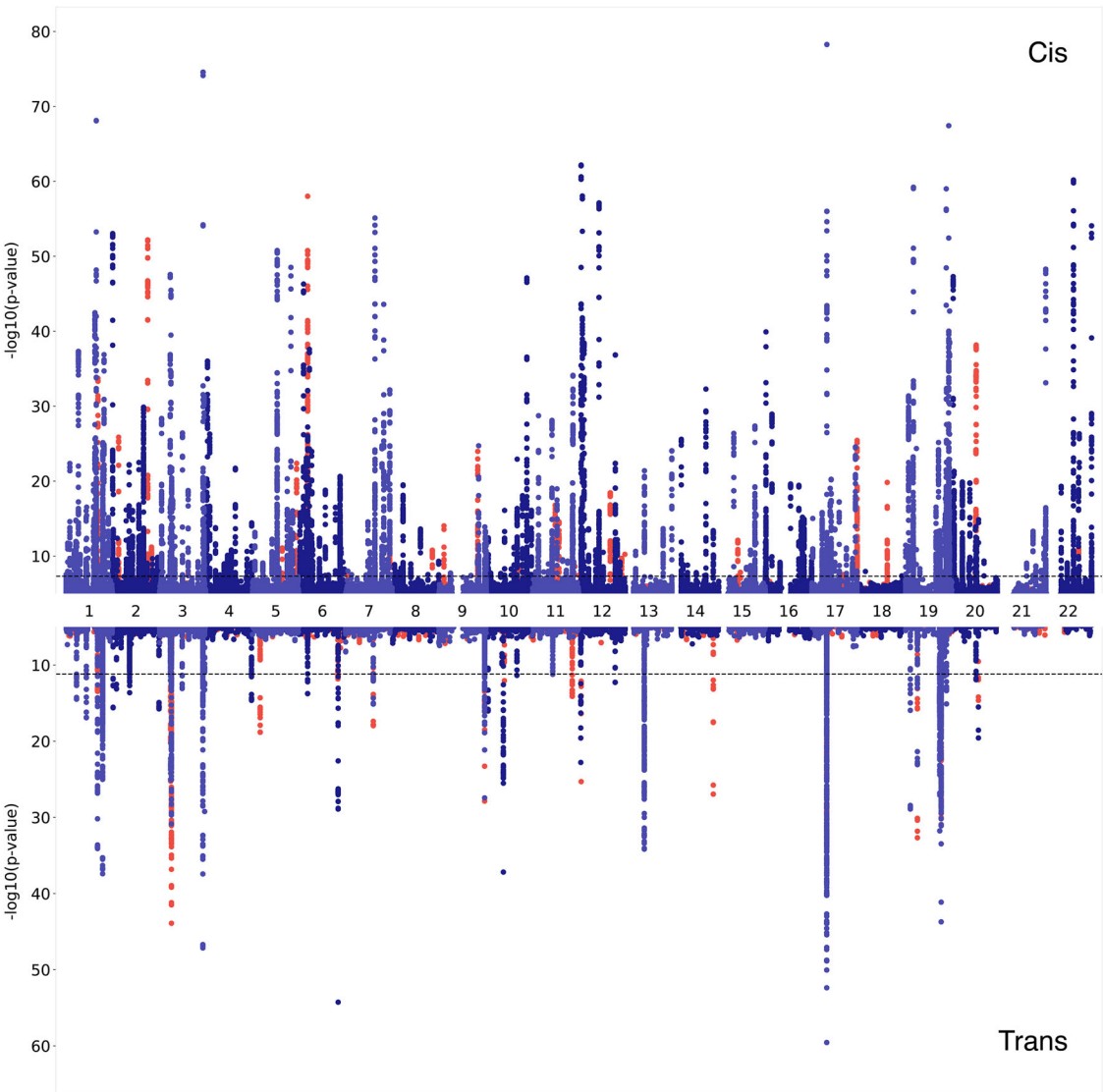

**Fig. 2 | Miami plot of the 455 proteins with genome-wide significant associations in this study.** In two shades of blue (denoting odd and even chromosomes) are the associations of proteins that have previously been reported in large-scale proteomics papers[10,14–18] (Supplementary Data 3 and 4). Proteins that were not found in aforementioned studies are coloured in red. *Cis* associations (upper part of the graph) are defined as being within 1 Mb of the transcription start site of the targeted protein, meanwhile all other associations are labelled as *trans* (lower part of the graph). Dashed lines represent the multiple-testing adjusted genome-wide significance thresholds, $p \leq 5 \times 10^{-8}$ for *cis* and $p \leq 6.6 \times 10^{-12}$ for *trans* associations.

Bayesian framework to assess whether the same underlying causal variant is responsible for the association with protein levels and the association with the outcome (disease risk). Of the 149 exposure (pQTL)–outcome (disease) pairs, 43 showed strong evidence of colocalisation (PPH4 > 0.8), suggesting direct genetic influences on disease via specific proteins, highlighting targets for future therapeutic intervention. For the remaining exposure-outcome pairs without strong colocalisation evidence, it is possible that multiple independent causal variants contribute to the outcome trait at the locus. Since the coloc.abf method assumes a single causal variant, this could result in an underestimation of colocalisation for loci with more complex genetic architectures, where true colocalisation may still exist. Notably, 14 out of the 17 proteins with statistically significant forward MR associations had at least one association passing this sensitivity test, providing further supporting evidence (Supplementary Data 6). Among the 43 colocalising protein-disease outcome pairs, a few of the most interesting associations will be discussed in greater detail (Table 1).

We conducted replication Mendelian randomisation (MR) analyses for 14 proteins associated with medical traits identified in this study. Replication was performed using either a fully independent exposure and outcome (Stage 1) or, in cases where no similarly powered outcome study was available, with an independent exposure and the same outcome as in the discovery MR (Stage 2). In total, we successfully replicated 11 protein–medical outcome associations in Stage 1 and another 10 in Stage 2, achieving replication for 21 out of 33 protein–outcome pairs with suitable instruments (Supplementary Data 7).

The genes harbouring each of the pQTL passing MR and sensitivity tests were checked in the genebass database of aggregate associations of rare variants, but no significant aggregate associations with any medical phenotype in the UK Biobank were found[21].

**B3GAT1 and prostate cancer**

We found that genetically decreased levels of B3GAT1 (CD57; beta-1.3-glucoronyltransferase 1) are associated with increasing risk for prostate cancer (MR effect size = −0.080, MR $p = 1.9 \times 10^{-7}$). The reverse MR is not significant and the coloc posterior probability H4 is 0.91 (Fig. 5A, B). Notably, the *cis*-associated rs78760579 (effect allele G, effect size = −0.80, $p = 3.7 \times 10^{-8}$) is in LD (Pearson $r^2 > 0.8$) with a recently reported variant, rs878987 ($p = 2.7 \times 10^{-6}$ in this study), detected as the lead variant in large case-control prostate cancer GWAS[22,23] ($p = 4.8 \times 10^{-8}$).

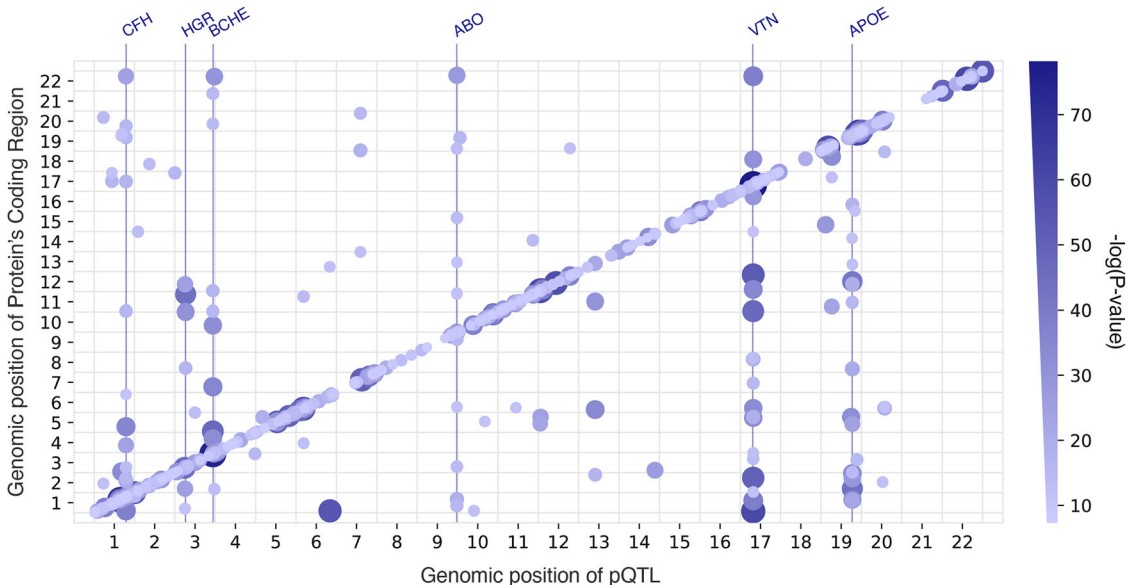

**Fig. 3 | pQTL distribution across the genome.** 505 pQTL are plotted against the locations of the genes encoding those protein targets. Both the size and the colour of the points show the significance of the genetic association. Pleiotropic regions (hubs) are marked with vertical lines and the genes underlying the associated regions are named at the top. Only the pleiotropic regions with at least 5 *trans* associated proteins are shown as hubs. pQTL protein Quantitative Trait Locus.

However, we were unable to replicate this finding independently using a non-overlapping study, likely due to the lower power of the only available prostate cancer GWAS (with 8.7 times fewer cases)[24]. Nonetheless, semi-independent replication was successful using the same outcome study along with two independent exposures (eQTL and SomaLogic pQTL; see Supplementary Data 7). This semi-independent replication supports the robustness of the observed association.

### LTK and diabetes

We have shown that rs1473781 is a *cis* pQTL (effect allele A, effect size = 0.69, $p = 7.75 \times 10^{-13}$) for LTK (leukocyte receptor tyrosine kinase). This SNP was shown to causally affect type-2 diabetes risk, mediated by LTK (MR effect size 0.054, MR $p = 6.4 \times 10^{-6}$), there is no effect in reverse MR, and the association passed the colocalisation sensitivity test with a posterior probability of colocalisation (H4) of 0.95 (Fig. 5C, D).

We could not replicate this finding as no QTL in any of the replication datasets reached the genome-wide significance threshold (Supplementary Data 8).

### NIF3L1 and macular degeneration

In the NIF3L1 (transcriptional activator NGG1 Interacting Factor 3 Like 1) GWAS, the *cis* sentinel SNP, rs10931931, (effect allele T, effect size = 0.87, $p = 6.3 \times 10^{-12}$) was found to be causally linked to macular degeneration and decrease its risk (MR effect size −0.11, MR $p = 2.0 \times 10^{-6}$), with no effect in reverse MR. This association passed the colocalisation test (H4 = 0.81).

However, the association could not be replicated in a semi-independent analysis using an eQTL dataset and the same macular degeneration GWAS (Supplementary Data 7). Although the replication test met the significance threshold, the effect sizes in the discovery and replication analyses were contradictory (replication MR effect size = 0.63, $p = 1.2 \times 10^{-4}$).

### Other notable associations

A *cis* sentinel SNP, rs13258747, (effect allele T, effect size = −0.62, $p = 1.7 \times 10^{-11}$) for *NTAQ1* (N-terminal glutamine amidase 1; also known as WDYHV1), showed an effect on the levels of testosterone (MR effect size 0.026, MR $p = 4.6 \times 10^{-7}$). However, we were unable to replicate this link using colocalising eQTL from eQTLGen[25] as an MR exposure with the same testosterone level outcome study (Supplementary Data 7).

Finally, we found a *cis* SNP, rs72941336, (effect allele T, effect size = 0.13, $p = 2.7 \times 10^{-15}$) for AAMDC (Adipogenesis Associated Mth938 Domain-Containing protein) to be involved with intrinsic epigenetic age acceleration related to DNA methylation (MR effect size 0.22, MR $p = 1.0 \times 10^{-5}$). This link was successfully replicated with SomaScan pQTL and eQTL instruments (Supplementary Data 7).

### Discussion

The application of broad-capture proteomic profiling and linking that to genomics holds great potential to increase our understanding of biology and the mechanisms underlying various diseases. In this study we present the results of one of the most comprehensive proteomic GWAS, encompassing 6432 blood plasma proteins of the SomaScan v4.1 assay, of which 1533 have not been measured in any large-scale proteogenomic study to date. A total of 505 pQTL were identified for 455 proteins, 76% of which were in *cis* (within 1 Mb of the gene encoding that protein). These results include unexplored associations with 58 proteins, 31 (53%) of which were categorised as *cis*. As for *trans* associations, we observed that 49% of them (60 out of 123) are linked to one of the *trans CFH*, *HRG*, *BCHE*, *ABO*, *VTN* or *APOE* hubs (Fig. 2 and Supplementary Data 1). Notably, this group of 6 genes has a marked enrichment in the regulation of coagulation and proteoglycan binding (Supplementary Data 9). A comparable enrichment pattern was also observed in another plasma proteomics study, utilising a different proteomics assay[26], though not noted in other tissues[27]. These findings suggest that these hubs may represent true biological effects or, alternatively, be artifacts related to the plasma sample preparation process.

Overall, we observed a higher proportion of *cis* signals compared to 10–31% in other large-scale proteomics studies[10,16] This is consistent with the understanding that smaller cohort sizes, such as the $n = 200$ in this study, are more likely to detect proteomic GWAS signals with larger effect sizes, which are predominantly *cis*[9]. Power analysis based on theoretical expectations supports this interpretation, indicating that the study is well-powered to detect relatively strong effects (Supplementary Fig. 4). Correspondingly, the sentinel pQTL identified in this study are linked to protein level changes of at least 7% (Supplementary Data 1). These findings underscore the role of sample size in the types of signals detected; while smaller cohorts like this one predominantly capture *cis* signals with large effect sizes, larger cohorts allow for the detection of more *trans* associations as the sample size increases beyond 5000[6].

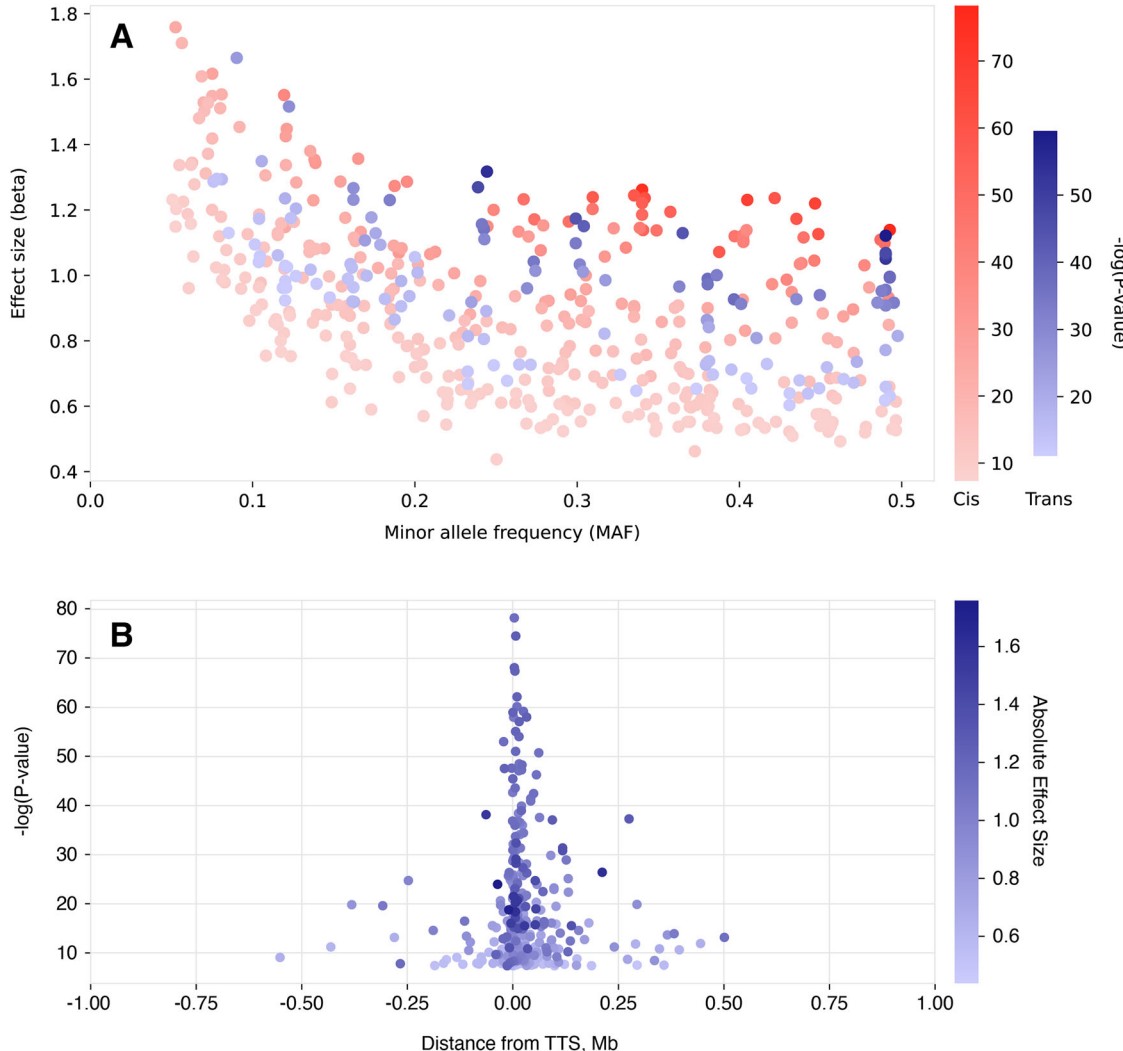

**Fig. 4 | Effect sizes and significance of *cis*-pQTL are influenced by allele frequency and genomic proximity to the transcription start site. A** SNPs with lower minor allele frequency tend to have larger absolute effect sizes. *cis* (in red) and *trans* (in blue) associations are coloured separately to show the same L-shaped distribution. *cis*-pQTL are both more detectable and have higher effect size at lower allele frequencies than *trans*-pQTL. Variants with minor allele frequency <0.05 were excluded from this analysis. Effect sizes (beta) were calculated on rank-based inverse normal transformed protein level data. **B** Distribution of *cis*-pQTL association *p* values by distance to the transcription start site of the gene encoding the protein. Closer to the TSS, pQTLs tend to be more significant, with a corresponding increase in effect size. TSS transcription start site, pQTL protein Quantitative Trait Locus.

Consistent with patterns noted in both molecular and whole-organism characteristics[28], we have identified an inverse correlation between allele frequency and effect size (Fig. 4A). Remarkably, this trend holds even though we omitted rare variants (MAF < 0.05) from our analysis, due to the constraints of a smaller sample size. Similarly, an inverse correlation between the *cis*-pQTL association strength (*p* value/ effect size) and distance to the transcription start site is observed (Fig. 4B), as in previous proteomics studies[9,26]. Notably, only 2 out of 382 *cis*-pQTL identified herein fell outside the 500 kb range of the transcription start site for that protein. This is in agreement with previous theoretical research[29] and suggests interpreting pQTL outside of this range as *cis* with caution. Finally, we have observed that when the pQTL were categorised based on their predicted protein-altering properties, there were significant differences in the explained variance of protein abundances. Specifically, the variants predicted to have a high or moderate impact played a more significant role in contributing to changes in protein abundance, consistent in both *cis* and *trans* associations (Supplementary Fig. 2A, B).

In our analysis, we have identified notable cases with FCGR2B and FUCA1 where other proteins (FCGR2A, FCGR2C, and FUCA2, respectively) exhibit extensive similarity in their amino acid sequences. This high level of homology can lead aptamers to bind incorrectly to proteins they were not meant to target, creating false or misleading associations. With 70.5% of human genes having at least one paralog, 71.4% of which are located within 1 Mb[30], future research in proteomics should exercise caution when interpreting results due to misidentification and amino-acid sequence similarity for both *cis* and *trans* associations[31].

We then followed up the 31 *cis*-pQTL for the hitherto unexplored protein group with Mendelian Randomisation, incorporating a reverse MR filter to address reverse causation. This enabled us to assess the potential causal role of the levels of these proteins in disease, uncover new biology and potential drug targets. After extracting traits categorised as medically relevant (see 'Methods', Supplementary Data 4, 5) from the OpenGWAS[19] database, we have identified 149 significant protein-outcome associations passing both forward and reverse MR tests for 16 distinct proteins. Of these, a total of 43 colocalising protein-outcome pairs for 14 proteins and 39 medically relevant outcomes were observed. We highlight 7 protein-disease associations with newly discovered pQTL. Of these, B3GAT1, LTK and NIF3L1 have been researched more thoroughly in pre-existing literature and are discussed in more detail here.

## Table 1 | Noteworthy *cis*-pQTL associations with colocalising Mendelian randomisation outcomes

| *cis*-pQTL | Gene | Colocalising MR outcome | Coloc PP.H4 | GWAS-log($p$) | GWAS effect size | MR-log($p$) | MR effect size |
|---|---|---|---|---|---|---|---|
| rs78760579 | B3GAT1 | Prostate cancer | 0.91 | 7.4 | −0.80 | 6.7 | −0.080 |
| rs1473781 | LTK | Type 2 diabetes | 0.95 | 12.1 | 0.69 | 5.2 | 0.054 |
| rs10931931 | NIF3L1 | Early age-related macular degeneration | 0.81 | 11.2 | 0.88 | 5.7 | −0.11 |
| rs13258747 | NTAQ1 | Total testosterone | 0.88 | 10.8 | −0.62 | 5.5 | −0.02 |
| rs72941336 | AAMDC | Intrinsic epigenetic age acceleration | 0.90 | 14.6 | −0.99 | 5.0 | 0.22 |
| rs1169084 | BCL7A | Systolic blood pressure | 0.85 | 9.7 | 0.68 | 5.5 | 0.23 |
| rs56953556 | COMMD10 | Parental longevity (mother's attained age) | 0.90 | 11.1 | −1.34 | 5.4 | −0.016 |

The table showcases causal associations identified in this study between pQTL, protein levels and diseases or health outcomes that may be of medical interest. *pQTL* protein Quantitative Trait Locus, *MR* Mendelian randomisation, *Coloc PP.H4* posterior probability of colocalisation. Proteins represented: *B3GAT1* Beta-1,3-Glucuronyltransferase 1, *LTK* leukocyte receptor tyrosine kinase, *NIF3L1* NGG1 interacting factor 3 like 1, *NTAQ1* N-Terminal Glutamine Amidase 1, *AAMDC* Adipogenesis Associated Mth938 Domain Containing protein, *BCL7A* BAF Chromatin Remodelling Complex Subunit BCL7A, *COMMD10* COMM Domain-Containing Protein 10.

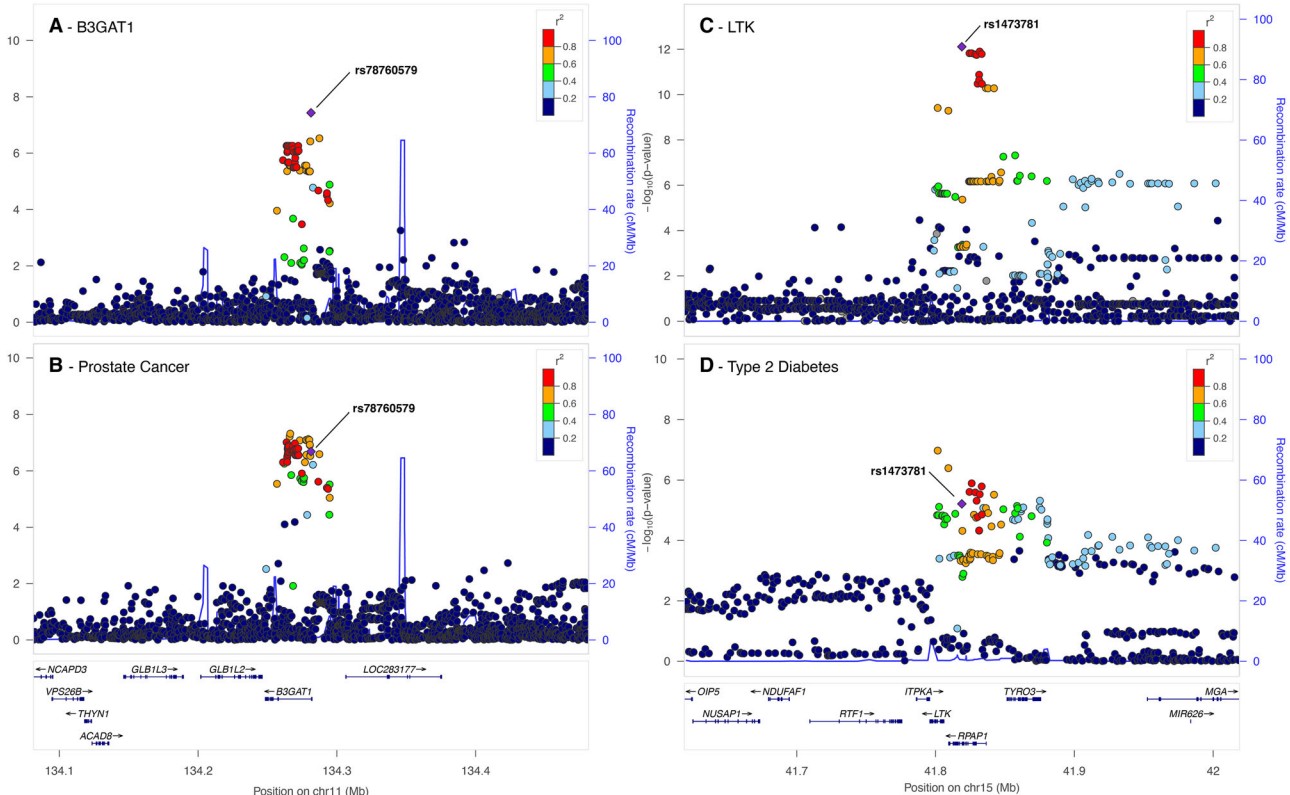

**Fig. 5 | LocusZoom plots showcasing notable potentially causal protein level and disease associations. A, B** B3GAT1 and prostate cancer. LocusZoom plots show a visual colocalisation comparison between the local association architecture for circulating B3GAT1 protein levels in our study (**A**) and that for ebi-a-GCST006085, a case-control prostate cancer study (**B**). The two studies colocalise in this locus with posterior probability of colocalisation (H4) = 0.91. **C, D** LTK and type 2 diabetes.

Colocalisation comparison between the local association architecture for circulating LTK protein levels in our study (**C**) and that for ebi-a-GCST006867, a case-control type 2 diabetes study (**D**). The two studies colocalise in this locus with H4 = 0.95. The legends indicate Linkage Disequilibrium (LD) patterns between the sentinel SNP and others in the region.

The B3GAT1 protein is an enzyme that participates in the biosynthesis of glycosaminoglycans—long, unbranched polysaccharides found on the cell surface and in the extracellular matrix that play a part in cell signalling[32] and adhesion[33]. B3GAT1 knockdown in human tissues and mice experiments have been previously shown to moderate glycosaminoglycan structure, inhibiting spreading of tumour cells and increasing the survival of the animals[32]. Other studies have shown that human prostate luminal cell tumours continue expressing B3GAT1 (CD57) upon turning malignant and that this differentiation is among the most common prostate cancer phenotypes[34,35]. Taken together, these results show that while B3GAT1 was

shown to be involved in prostate cancer, this is the first time the causal role of the protein in the disease has been suggested.

The LTK (Leukocyte Tyrosine Kinase) protein is a receptor tyrosine kinase that belongs to the insulin receptor superfamily. Its specific function is not fully understood, but it is believed to play a role in neuronal development, immune response and cancer[36]. LTK variants also have evidence of being involved in lupus erythematosus, an autoimmune disease[37,38]. Despite its name, LTK is primarily internally expressed in adipocytes[39]. Annotation of LTK via STRING[40] for protein-protein interactions has elucidated two potential pathways through which LTK may be participating in Type 2

diabetes. PIK3R1 (phosphoinositide-3-kinase regulatory subunit 1), a protein facilitating insulin signal transduction, mutations of which have been shown to trigger insulin resistance[41], has been shown to be essential in for LTK signal transduction through co-immunoprecipitation[42]. In contrast, IRS1 (insulin receptor substrate 1), a protein in which mutations are also well-documented to lead to type 2 diabetes[43], does not have biochemical data on interactions with LTK in humans. Instead, they are linked through text mining[38]. Finally, LTK was found to be associated with asthma, dermatitis, and cardiac arrhythmia in the FinnGen database, but not with type 2 diabetes[44].

NIF3L1 (NGG1 Interacting Factor 3 Like 1) is a little-studied protein that is broadly expressed in most tissues, including the retina[45], and involved in transcriptional regulation[39]. A recent study utilising single-cell RNA sequencing has highlighted *NIF3L1* as actively transcribed across multiple retinal cell types[46]. However, their exclusive investigation into the noncoding region was unsuccessful in identifying a causal variant responsible for risk of age-related macular degeneration or any other eye disease tested. The complex Linkage Disequilibrium pattern observed (Supplementary Fig. 5) encompasses 7 other genes and further research is necessary to see if the sentinel variant rs10931931 identified in this study is indeed driving the causal relationship between NIF3L1 and age-related macular degeneration.

As for drug repurposing, 7 out of 14 proteins identified to be linked to a medical trait in this study have approved or investigational drugs in DrugBank as of April 2025 (Supplementary Data 10)[47]. With half of the proteins having no entry in the database, none of the existing approved indications match the ones proposed in this study, showing truly novel therapeutic opportunities previously unexplored in clinical development. This illustrates the utility of pQTL in both discovering potential new drug targets and reimagining existing targets for different diseases.

Only a single other GWAS study has used the SomaLogic v4.1 panel, to date. The study focused on 466 chronic kidney disease patients of African American descent, performing Mendelian Randomisation on estimated Glomerular Filtration Rate and colocalisation with 778 phenotypes from the UK Biobank[8]. Of the 31 proteins selected for MR in our study, 13 (42%) had sentinel SNPs in high LD ($r^2 > 0.6$) with those in VIKING, highlighting shared genetic signals. Additionally, we found that 58% of all the *cis* pQTL identified in this study are in high LD ($r^2 \geq 0.6$) with those reported in the referenced one, highlighting both shared and ancestry-specific effects of genetics on protein levels (Supplementary Data 2). In contrast to the previous study, we have analysed a healthy European population and have broadened our research to encompass a comprehensive range of 3772 medically relevant traits available in the OpenGWAS database.

To summarise the *cis* pQTL replication results, 353 out of 382 *cis* associations were successfully replicated in either Pietzner et al. or Surapaneni et al. For 29 *cis* pQTL corresponding to 28 proteins, no suitable replication dataset was available. However, given the high replication rate observed with existing published data, we anticipate strong replication in future studies utilising this proteomics platform.

We attempted to replicate the potential causal links identified through Mendelian Randomisation using two approaches: by leveraging fully independent exposure datasets and by employing independent outcome GWAS where available. While 26 out of 39 replicable protein-disease associations were successfully validated, a third did not replicate (Supplementary Data 5).

Notably, 10 of the 13 non-replicated associations were related to blood cell counts. This suggests that protein-blood cell associations may be more challenging to replicate, possibly due to the complexity and variability of haematological traits across different cohorts and genetic ancestries. Additionally, the lack of replication could be due to several factors, such as lower statistical power in the available outcome GWAS, especially when the study sample sizes were smaller or had fewer cases. Differences in study design, cohort demographics, or instrumentation platforms (e.g., SomaLogic vs. Olink vs. eQTL)[48,49] might also introduce variability that hampers replication.

The main limitation of this study is its relatively low sample size, with proteomic profiling being performed in 200 individuals. While the sample size falls short of the standard typically required for GWAS with effect sizes of organismal-level traits[28], it has proven sufficient for analysing molecular traits, when variance in protein levels is low, such the 5% observed in the SomaScan platform[50]. Our power analysis demonstrates that with a sample size of 200, we achieve >80% power to detect genetic variants explaining ≥20% of variance in protein levels (Supplementary Fig. 4). In practice, we identify pQTL explaining as little as 7% of the protein level variance, alongside large effect sizes. This is consistent with prior molecular phenotype studies, where maximum trait variance explained by the QTL reached 22–39%[28,51,52]. Conversely, other proteomics studies using SomaLogic technology, albeit with a much larger sample size ($n > 10,000$), were similarly able to uncover pQTL that explained up to 75% of the variance in the observed protein levels[16].

Clearly, an increased sample size would enable discovery of variants with smaller effects. For instance, in this study only 4.6% of the proteins had a genome-wide significant *cis* association, while the largest proteomic studies report >90%[13]. Increasing the sample size would help not only with identifying smaller *cis* associations, but would also allow detection and pathway analysis of more *trans* associations, as their number increases and exceeds that of *cis* associations as the study power grows[14]. A better powered proteomics association study would in turn increase power in the downstream MR, allowing further causal protein-disease connection discovery. Nevertheless, studies of this size ($n = 200$) are manifestly able to identify the strongest proteomics MR instruments. This approach to proteomics may serve as an alternative, yet complementary strategy to large-scale studies toward cost-effective instrumentation of more human proteins.

In this study, we have restricted our MR instrument selection to proteins that were not assayed previously in large-scale SomaLogic or Olink studies. However, we acknowledge that this definition is subject to change as new versions of proteomics assays are released and new studies are conducted using them. One such instance relevant here is the publication of the results of the Olink Explore 3072 assay in the UK Biobank[6] in-between our MR instrument selection and the completion of this study. Of the 31 proteins that we at the time annotated as previously unstudied in large-scale proteomics GWAS, 10 were found to be measured in the Olink Explore 3072. Among these, 4 proteins were included in our final set of 14 with potentially causal links to medical traits. Given that SomaLogic and Olink assays do not always capture the same signal[49], we assessed colocalization of the *cis* genetic signals for these proteins. Indeed, of the 4 matched protein measurements, only one genetic signal colocalised (Supplementary Data 8).

The other limitation of the study is that we focused exclusively on the common variation in the genome (MAF > 0.05), again due to sample size. As shown in Fig. 3a, rarer variants are more likely to have a larger effect on protein levels, which may translate into a greater influence on disease risk.

Finally, the sample used in this study is of exclusively European heritage, representing the predominant ancestry of populations where the majority of current discoveries have been made. As is the case with disease studies[53], deploying proteomic GWAS to populations of diverse continental ancestries will reveal further components of the genetic architecture, due to the different variants segregating. Indeed, our analysis suggests that proteomic GWAS of even relatively modest sample sizes from diverse populations may be a fruitful strategy to increase the number of proteins that can be used as instruments for MR.

Our findings provide strong support for continuing to increase the number of proteins under study in genome-wide association, so that many hitherto unstudied proteins will have genetic evidence available in drug development pathways. We further show that studies of modest sample sizes can reveal highly significant, novel pQTL. Deployment of this approach across multiple ancestries may be a cost-effective way to maximise the number of proteins for which genetic instruments are available. Finally, we identify new connections between proteins and disease risk which illuminate mechanisms and will help pave the way for new or repurposed therapies.

## Methods

### Study participants

The Viking Health Study—Shetland (hereafter VIKING1) is a geographically defined cohort with grandparents from the Shetland Isles, north of Scotland, which seeks to identify genetic factors influencing cardiovascular and other disease risk[54]. High levels of historical endogamy are reflected in the distinct gene pool of the VIKING1 cohort, as indicated by both common and rare genetic variants that set it apart from the rest of the British Isles and Europe[55,56]. Recruitment of 2105 volunteers took place between 2013–2015. Each participant completed a health survey questionnaire and attended a 2-h measurement clinic. Following that, overnight fasting blood samples were collected and frozen for downstream analyses. All participants gave informed consent, and the study was approved by the Southeast Scotland Research Ethics Committee, NHS Lothian (reference: 12/SS/0151). All ethical regulations relevant to human research participants were followed.

A subsample of 200 participants was chosen for this study, with all 4 grandparents originating from the Shetland Isles and with minimal kinship to the rest of the $n = 200$ sub-cohort. The highest genomic relatedness in this sub-cohort was 6%. Ages ranged from 19–91 (mean 52.6, s.e. 16.0), with females composing 53.5% of the sub-cohort (Supplementary Data 11).

### Plasma samples and protein measurement

Following standard processing protocols (clotting, centrifugation and aliquoting), EDTA-treated fasting blood plasma samples were immediately frozen at $-40\,°C$ and thereafter kept at $-70\,°C$ for long-term storage. Frozen aliquots (500 μl) were shipped on dry ice to SomaLogic Inc. (Boulder, Colorado, USA) for proteomic analysis. All 200 blood plasma samples were measured with the SomaScan assay, version 4.1 (SomaLogic Inc.). This platform employs chemically modified DNA aptamers known as SOMAmer reagents, which are short single-stranded DNA oligonucleotides incorporating hydrophobic nucleotide modifications that enhance protein-binding diversity and affinity. The reagents are selected via a proprietary SELEX process against proteins in their native form, with high specificity for intact tertiary structures[57].

The assay is specifically designed for human plasma analysis and measures protein levels using 7596 aptamers, covering 6432 unique human protein targets across a dynamic concentration range of 10 orders of magnitude. SomaScan assay includes several steps: (1) incubation of three levels of plasma dilutions (20%, 0.5%, 0.005%) with bead-immobilised SOMAmers, (2) protein capture and labelling with NHS-biotin, (3) selective release and enrichment of SOMAmer–protein complexes, and (4) SOMAmer quantification via hybridisation to complementary DNA on microarrays. Protein concentrations are measured in relative fluorescent units (RFU) that are proportional to the actual amount of target protein epitope in the plasma sample[57].

### Data quality control

Quality control of the assay results was performed both by SomaLogic[58] and using in-house methods. In brief, SomaLogic quality control followed their validated quality system and included experimental controls in each 96-well plate: five pooled Calibrator Control replicates, three pooled Quality Control (QC) replicates, and three buffer (no-protein) controls. Normalisation steps include hybridisation control normalisation using non-protein-binding control oligonucleotides, intra-plate calibrator adjustment matching a global reference, plate scaling and SOMAmer-specific calibration to remove between-plate variability, and median normalisation to a healthy pooled reference to reduce sample-to-sample variation. Calibration and QC replicates were used to correct systematic variance, with performance assessed against proprietary SOMAmer-specific acceptance thresholds[58]. This process by SomaLogic altogether marked 289 aptamer measurements as inconsistent.

We performed further quality control based on the overlap (>5%) of observed signal data points between buffer control (plasma-free) and the actual samples. This may indicate a lack of sensitivity or specificity in the aptamer binding, considering that most (7526 out of 7596) aptamer signals do not overlap with the buffer control signal. Aptamers were also flagged for targeting non-human proteins or having no specified target at all. A total of 595 aptamers were marked this way, with an overlap of 33 with the SomaLogic quality control. The flagging system was only used descriptively to track the robustness of downstream analysis, without filtering any aptamer measurements. Protein abundances were then filtered by removing outliers outside the three interquartile range from the median raw measurement of each protein level. This resulted in GWAS having differing sample sizes with a median of 198 (min 174, max 200).

### Protein-phenotype and technical covariate associations

Confounding factors are variables that can influence measured protein levels in the samples. These can be either inherent (e.g. sex, age), or technical (e.g. blood plasma sample storage time or co-ordinates on measurement plate). Technical artifacts, such as batch effects and some confounding factors can be accounted for to improve power and decrease the risk of false associations[59,60]. For each aptamer, we performed multiple regression with the following covariates: biological sex, age, sample storage time in the freezer, season of the year when the plasma sample was taken, 96-well measurement plate number, row, and column. Forward stepwise selection with a likelihood ratio test (scipy 1.9.1, python 3) was performed to determine the influence of these covariates on each of the measured aptamer levels. All previously described covariates, except for sampling season had a statistically significant ($p < 0.05/7596 = 6.58 \times 10^{-6}$, Bonferroni corrected for 7596 aptamers) effect on at least one of the measured protein levels. Therefore, all covariates except for sampling season were included in the GWAS model as fixed effects.

Similarly, principal components (PC) of the Genomic Relatedness Matrix (GRM) were used to correct for population stratification[61]. The GRM was created, and PC1-20 extracted for the whole 2005 individuals in the VIKING1 cohort using PLINK[62]. PCs were computed using parameters --maf 0.0025 and --nonfounders for autosomal variants.

For the 200 individual sub-cohort with proteomic data, the PC1-20 were analysed using multivariate regression one at a time via nested models for each aptamer. No principal components were significant ($p < 0.05/(7596*20) = 3.29 \times 10^{-7}$, Bonferroni corrected for 7596 aptamers and 20 PCs) for all aptamer measurements and most of the aptamers exhibited a unique combination of significant PCs. However, due to the small sample size and limited degrees of freedom, in addition to the Scree plot having a clear Inflection Point (Supplementary Fig. 6), we decided to include only PC1-3 for all aptamers. As further post-GWAS analysis showed, this controlled for population stratification with a median genomic inflation control factor $\lambda = 1.005$, s.e. 0.015 (min 0.945, max 1.127) across all GWAS. $\lambda$ of 44 out of 7596 GWAS performed in this study fell outside the accepted 0.95–1.05 range, with 2 yielding genome-wide significant pQTL. These pQTL were not considered for further post-GWAS analysis.

### Genotyping and imputation

Individuals were genotyped using the HumanOmniExpressExome8 v1-2_A (Illumina) platform. Data was called with Beadstudio-Gencall v3.0 (Illumina). SNP genotype quality control (QC) was carried out using PLINK 1.9[62]. Only high-quality variants were selected: those with Hardy-Weinberg Equilibrium test $p > 1 \times 10^{-6}$, SNP call rate >98%, individual call rate >97%. In addition, we detected and removed Mendelian errors by using cohort pedigree information and removed monomorphic SNPs. After initial Quality Control, 611,836 autosomal SNPs remained in the dataset. We then imputed SNPs to the Haplotype Reference Consortium (HRC) panel v1.1 using the Sanger Imputation Service[63]. Imputed variants with low imputation quality scores (INFO < 0.4) were removed prior to downstream analysis. This resulted in 5,462,684 common autosomal variants with a minor allele frequency (MAF) of ≥0.05 in the VIKING sub-cohort of 200 individuals with proteomics data reported in this study.

## Genome-wide association study

Due to skewness in the distributions, the relative protein detection levels of all 7596 aptamers were independently rank-based inverse normal transformed prior to GWAS. GRAMMAR residuals were computed by first regressing out the fixed effect covariates: sex, age, the previously described technical covariates and PC1-3 of the genetic relationship matrix (Supplementary Fig. 6), before modelling the relationship matrix as a random effect (GenABEL[64]). The resulting GRAMMAR residuals were then divided by the Gamma factor (GRAMMAR-Gamma)[65] and tested against genotypes using RegScan v0.5[66].

Following the GWAS, variants with a minor allele frequency (MAF) < 0.05 were excluded due to the limited sample size.

## Gene enrichment analysis

Gene enrichment analysis was conducted using the PANTHER 19.0 Overrepresentation Test[67,68] with the GO Ontology database, version 2024-06-17[69,70]. The analysed dataset contained *trans* hub genes identified in this study (*CFH*, *HRG*, *BCHE*, *ABO*, *VTN* and *APOE*), compared against a reference list of 6432 human genes present in the SomaLogic v4.1 assay, of which 6278 mapped to the GO Ontology database. Fisher's Exact Test was utilised to identify significant overrepresentations, with False Discovery Rate (FDR) correction applied for multiple testing adjustments (Supplementary Data 9).

## *cis*- and *trans*-associations

An association was defined as *cis* if the associated SNP was within 1 Megabase (Mb) of the transcription start site of the gene encoding the protein that was targeted by that aptamer. Conversely, associations found outside this region or on another chromosome were defined as trans associations. To assess the proximity of a particular pQTL to the gene encoding the protein, we extracted the transcription start sites for the aptamer protein targets using Ensembl Biomart (build GRCh37), accessed in July 2022[71]. We used two different thresholds to define significance of the association: $5 \times 10^{-8}$ for *cis*-pQTL, where there is prior expectation of an association, and $6.58 \times 10^{-12}$ ($5 \times 10^{-8}/7596$, number of aptamers) for trans associations.

Independent associations and LD proxies. To identify independent association signals (sentinel SNPs) we used clumping, as implemented in PLINK 1.9[62], with a window of ±250 kb around the significant variants, and LD Pearson $r^2 < 0.01$ against a reference panel of a random subset of 10,000 unrelated genomically British individuals from UK Biobank[72,73]. PLINK options used were –clump-kb 250 --clump-r2 0.01 --clump-p1 0.00000005 --clump-p2 0.0000025.

For proteins with multiple genome-wide significant *cis* associations, further filtering was performed since some genomic regions have long-range Linkage Disequilibrium (LD) patterns. Such associations were not considered independent if their clumping windows overlapped and only the SNP with the lowest *p* value was retained.

To assess how much individual independent pQTL contribute to differences in protein levels, we used the formula:

$$Variance\ explained = 2pq\beta^2$$

where:

$p$ is the major allele frequency,
$q = 1 - p$ is the minor allele frequency,
$\beta$ is the estimated effect size (beta) on the protein levels, as determined from the GWAS analysis.

The LDproxy[12] API was then used to define LD proxies in the regions of the genome-wide significant results. Proxies were selected if they were in LD with associated variants, using European 1000 Genomes Project populations (CEU, TSI, GBR, IBS), with Pearson $r^2 > 0.8$ within a 1 Mb window. LDproxy and its associated databases were accessed in August 2022.

pQTL and their linked proxies were annotated using Variant Effect Predictor (VEP) for their consequences on the protein structure[71]. The

consequences were divided into three categories—High, Moderate, and Low-Modifier. Only the most severe consequence(s) for each SNP was retrieved from the database and only the most severe consequence category was retained for the SNPs in LD with the sentinel variants in Supplementary Data 1.

The genome-wide significant results were then assessed for novelty. Proteins targeted in the most comprehensive proteomics studies using the SomaLogic v4.0 protein assay were not considered novel. Proteins from the Olink Explore 1536 panel were also treated as non-novel[14].

Subsequent analyses were only performed on the proteins that were not reported in the largest published proteomics study using the SomaLogic v4.0 assay[16], were not present in the Olink Explore 1536 panel and had at least one genome-wide significant *cis* signal in our study (Supplementary Data 4).

## *cis*-pQTL replication

To validate our *cis* pQTL findings, we retrieved full GWAS summary statistics from Pietzner et al.[10], where proteins were measured using the SomaScan assay version 4.0, in over 10,000 people from the Fenland cohort[10]. Ten proteins absent in the Pietzner et al. version of the assay but measured in other large-scale SomaLogic or Olink 1536 Explore studies[9,14,16,17] were excluded from both replication and subsequent post-GWAS analyses to preserve dataset integrity. This was to ensure consistency across datasets and avoid introducing variability arising from differences in assay technologies and study designs.

Six protein measurements were excluded because their associated aptamers are annotated with multiple protein targets or protein complexes, making it challenging to attribute the signal to a single protein. To ensure consistency and clarity, we restricted the replication efforts to aptamers targeting a single protein.

For six proteins targeted by multiple aptamers, we selected the aptamer best matching the effect size estimate observed in VIKING. Finally, in one instance the sentinel pQTL reported in this study was not genotyped in the cited study, leading to a replication dataset of 272 proteins with *cis* pQTL of the total 338 reported in this study. A *cis* pQTL was considered replicated if it met genome-wide significance ($p \le 5 \times 10^{-8}$) in the replication study for the same protein and exhibited the same effect size directionality as reported in this study.

In addition, we also assessed the replication of our *cis* pQTL in, at the time, the only other published SomaLogic v4.1 assay[8]. Surapaneni et al. quantified proteins in 466 chronic kidney-disease patients of African American ancestry. Due to differences in ancestry and the disease-based nature of their cohort compared to our general population cohort, we used this study as a secondary source for replication. Therefore, we examined whether the sentinel *cis* pQTL reported in the cited study were in LD ($r^2 > 0.6$) with the sentinel pQTL identified in this study. Pairwise LD between the sentinel pQTL from both studies was calculated using a reference panel of 10,000 unrelated, genomically British individuals from the UK Biobank[72,73] (Supplementary Data 2).

## Mendelian randomisation

Bidirectional two-sample Mendelian randomisation (MR) was performed to assess potentially causal associations between proteins (using *cis* sentinel SNPs as instrumental variables) and diseases and risk factors from the OpenGWAS[19,74] database. The MR was performed using the TwoSampleMR (0.5.6) R package[74], with the Wald ratio method. TwoSampleMR proxy search was enabled if the sentinel SNP in our study was absent in the referenced studies with default parameters (1000 Genomes reference, rsq = 0.8, palindromes = yes, maf_threshold = 0.3). In the two cases when the TwoSampleMR integrated proxy search failed to find a proxy for the sentinel SNP, the next strongest association within LD $r^2 > 0.8$ from our GWAS was supplied to the pipeline instead. Due to their complex LD structure which makes it difficult to annotate an association as *cis* or *trans*, we excluded SNPs that fall within the *ABO* (build GRCh37 chr9: 136.1311–136.1506 Mb) and *HLA* (build GRCh37 chr6: 2.9645–3.3365 Mb) regions[75]. Only very strong (*F*-statistic >30) MR

instruments were used in this study. Wald Ratio MR instrument strength was assessed as the $F$-statistic with the formula[76]:

$$F \text{ statistic} = (n - 2)R^2/(1 - R^2)$$

where:

$n$ is the GWAS sample size,

$R^2$ is the variance explained, or the independent pQTL contribution to differences in protein levels.

A subset of OpenGWAS[19,74] datasets was used: ieu-a, ieu-b, ebi-a, ukb-b (database accessed April 2023). These 4904 datasets were further filtered for medically relevant traits and outcomes by employing a large language model (LLM), ChatGPT 4 (version July 20). The LLM was asked to categorise each outcome in the OpenGWAS database on whether they are of medical importance. After five repeats, the resulting categorisation (yes/no/ambiguous) for each trait was assigned a numeric value (1, 0 and 0.5, respectively). If the five-round sum was 2.5 or higher, the corresponding trait was designated as exhibiting medical relevance and was selected for further study. The traits with resulting scores of less than 2.5 were manually curated for medical relevance before being discarded from the study.

To ensure the highest quality data for our MR analyses, we also implemented rigorous manual curation of OpenGWAS studies. Through the following steps, we removed 300 studies, resulting in 4604 high-quality datasets:

- Removed 4 specific studies known to show unreliable MR associations (GCST007800, GCST007799, GCST007236, ieu-b-5070);
- Excluded non-specific traits labelled as "none of the above";
- For bilateral measurements (e.g., left/right body parts), retained only right-side measurements to avoid redundancy;
- Harmonised naming conventions across studies (e.g., consolidated "total cholesterol levels" and "cholesterol, total" → "total cholesterol");
- Preferentially selected matching European ancestry studies with the largest sample or case count when the same trait was examined in multiple studies.

Combining the two filtering steps resulted in 3772 outcomes of perceived medical importance which we then used for MR (Supplementary Data 12 and 13).

First, we performed forward MR, with protein levels being used as exposures and disease and risk factors as outcomes. To distinguish causal effects from reverse causality, the significantly associated (FDR alpha threshold <0.01; equalling $p < 1.67 \times 10^{-5}$ in this study). We applied the False Discovery Rate (FDR) correction instead of the Bonferroni correction, as the latter is overly stringent when tests are not fully independent, increasing the chance of false negatives. Given the interdependence of both our instrumental variables (proteins) and outcomes (medical traits), we opted for a balanced approach using FDR correction with a more conservative alpha significance threshold of 0.01.

A total of 231 protein-disease pairs from the forward MR were used for reverse MR[77], where disease/risk factors were used as exposures and protein levels as outcomes using the "extract_instruments" function of TwoSampleMR[74]. We considered there to be no evidence for reverse causality if the reverse MR association was non-significant ($p > 0.01$).

To further validate our main findings, we performed replication MR for 14 proteins associated with medical traits identified in this study. Where possible, we attempted to replicate MR findings using both independent instrument and outcome datasets (Stage 1 replication). However, given the novelty of the SomaScan platform, we could not find a replication instrument dataset for all our proteins. In such cases we would use the same instruments as used in the discovery analysis, but instead used an independent outcome data for replication. Likewise, in some cases we could not find an outcome replication dataset of size comparable to the discovery study. In such cases we would use for replication an independent instrument dataset, but the same outcome dataset as used in the discovery MR analysis. The latter two were considered as a Stage 2 replication.

The following data were used for replication instrument datasets: (1) SomaScan v4.1 dataset from a disease-cohort of distinct ancestry (13 matched protein measurements)[8], (2) Olink Explore 3072 from the UK Biobank (4 matched proteins)[6], and (3) eQTLGen, using gene expression as a proxy for protein levels (13 matched proteins)[25]. While the pQTL and eQTL for the same protein often do not colocalise[10], the eQTLGen dataset has the great advantage of providing data for gene expression for almost all genes in the human genome.

Using the same approach as in the discovery analysis, to define independent instruments for each protein we performed clumping using the clump_data function from the TwoSampleMR R package[74] within each dataset's *cis* region, using the appropriate 1000 Genomes population reference (EUR or AFR), and by setting clump_kb = 1,000,000 and clump_r2 = 0.001.

For replication outcome datasets, we searched OpenGWAS[19], GWAS Catalog[78], and Google Scholar (July 2024) to identify additional GWAS studies of exact or similar traits. When an independent outcome study (i.e., one with no cohort overlap between studies) with comparable or larger sample size was unavailable or the Stage 1 fully independent MR analysis was not significant, we performed a semi-independent MR reusing the same outcome study as in the initial discovery MR, but with independent instruments (Stage 2).

To control for multiple comparisons, we first grouped discovery MR outcomes by similar traits (e.g., body fat percentage & BMI) or synonymous measurements (e.g., FVC & forced vital capacity) linked to the same protein. The Bonferroni-corrected significance threshold for both independent (Stage 1) and semi-independent (Stage 2) MR replications was $p = 0.05/(26 \text{ Stage } 1 + 80 \text{ maximum Stage 2 tests}) = 4.72 \times 10^{-4}$ (Supplementary Data 5 and 7).

We then performed colocalisation between the independent instruments and the VIKING cohort instruments (Supplementary Data 8), as well as the chosen replication outcomes (Supplementary Data 14). Unless otherwise specified, our MR and colocalisation replication procedures followed the protocols detailed in the discovery MR and colocalisation sections. An MR result was considered replicated if it demonstrated a consistent direction of MR effect between the discovery and replication analyses and had statistically significant evidence of association between at least one exposure and the outcome in the replication dataset.

### Colocalisation

Robust associations passing the bidirectional MR sensitivity test were then tested for colocalisation using the R coloc 5.1.0 package[20,79]. This method is used for analysis of two potentially related traits or diseases to investigate whether they share common underlying causal genetic variant(s), based on shared local genetic architectures of association. It involves testing five hypotheses: $H_0$ (no causal variants for either trait), $H_1$ and $H_2$ (causal variant for one trait only), $H_3$ (two independent causal variants, one for each trait), and $H_4$ (a single shared causal variant influencing both traits).

A 300 kb window around each sentinel SNP was selected for the test with default priors using the "coloc.abf" function. Default package prior probabilities (priors) were used, with Hypotheses 1 and 2 being assigned $1 \times 10^{-4}$ and Hypothesis 4 $1 \times 10^{-5}$. MAF unfiltered summary statistics were used for the colocalisation tests. Colocalisation was declared for tests for which the posterior probability of colocalisation ($H_4$) > 0.8.

### Association annotation

Clinically important associations passing all sensitivity tests described previously were manually assessed for recapturing known biology. The databases and tools used include the text-mining DISEASES platform[38], The Human Protein Atlas proteinatlas.org[80], GWAS and functional genomics database Open Targets Genetics[81,82], drug databases DrugBank[47] and ChEMBL[83], and the protein-protein interaction network database STRING[40].

Protein sequence alignment was investigated using clustalo v1.2.4[11].

Cumulative impact of multiple rare genetic variants for the protein and a linked disease outcome was investigated using genebass (gene-based

association summary statistics)[21]. Burden (overall burden of multiple variants) and SKAT-O (incorporates weighting for rare variants) test outcomes were checked for phenotype-wide significant associations in all Burden sets —putative loss of function, missense and synonymous.

## Statistics and reproducibility

Full descriptions of each analytical step are provided in the relevant sections of the 'Methods'. Below is a general summary of the statistical approaches and reproducibility considerations applied throughout the study.

The sample size for proteomics ($n = 200$) was determined based on cohort availability and ancestry constraints. No repeated protein quantification assays were performed, and all measurements were obtained from unique biological samples. Protein measurement data were rank-based inverse normal transformed prior to association testing. For Mendelian randomisation, only strong instruments (F-statistic >30) were used. The full dataset was analysed without exclusion except where detailed in individual method sections.

## Reporting summary

Further information on research design is available in the Nature Portfolio Reporting Summary linked to this article.

## Data availability

The summary association statistics for all proteomic GWAS in this study have been deposited to the GWAS Catalog (https://www.ebi.ac.uk/gwas/, Study Accession IDs GCST90436912–GCST90444200). All genetic polymorphisms analysed and reported are previously known and present in public variant databases, including those used for imputation (HRC v1.1). Numerical source data underlying the main and Supplementary Figs. are included in the Supplementary Data files or the deposited summary statistics. There is neither Research Ethics Committee approval, nor consent from individual participants, to permit open release of the individual-level research data underlying this study. The datasets generated and analysed during the current study are therefore not publicly available. Instead, the research data and/or DNA samples are available by managed access from accessQTL@ed.ac.uk, following approval by the QTL Data Access Committee and in line with the consent given by participants. Each approved project is subject to a data or materials transfer agreement (D/MTA) or commercial contract. The UK Biobank genotypic data used in this study as a LD reference panel were approved under application 19655 and are available to qualified researchers via the UK Biobank data access process.

## Code availability

All analyses were conducted using publicly accessible software tools, which are detailed both in the main text and within the 'Methods' section. Data handling was done in Python 3. Main modules used include pandas (v1.4), scipy (v1.4), numpy (v1.20) for data transformation and statistical analysis, requests (v2.22) for data download via API, and matplotlib (v3.2) for creating graphs. The data analysis pipeline and other scripts used in this study are available on GitHub https://github.com/viking-genes/SomaScan-pQTL.

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

## Acknowledgements

The Viking Health Study—Shetland (VIKING1) was supported by the MRC Human Genetics Unit quinquennial programme grant "QTL in Health and Disease" (U. MC_UU_00007/10). DNA extractions and genotyping were performed at the Edinburgh Clinical Research Facility, University of Edinburgh. J.K. acknowledges the MRC Doctoral Training Programme in Precision Medicine (MR/N013166/1). L.K. was supported by an RCUK Innovation Fellowship from the National Productivity Investment Fund (MR/R026408/1). P.N. was supported by UKRI's Medical Research Council (MC_PC_U127592696, MC_PC_U127561128 and MC_UU_00007/10) and the BBSRC (BBS/E/RL/230001A). We would like to acknowledge the invaluable contributions of the research nurses in Shetland, the administrative team in Edinburgh and the people of Shetland. For the purpose of open access, the author has applied a Creative Commons Attribution (CC BY) licence to any Author Accepted Manuscript version arising from this submission.

## Author contributions

Conception and design: J.F.W. and L.K.; Data curation and analysis: J.K.; Scripting: J.K., P.T., and L.K.; Writing: J.K.; Review and editing: J.K., J.F.W., L.K., P.N., and P.T.

## Competing interests

P.T. and L.K. are currently employed by and have share options in BioAge Labs. The remaining authors declare no conflicts of interest.
