## [Transparent Peer Review file · Communications Biology]

Efficient candidate drug target discovery through proteogenomics in a Scottish cohort

Corresponding Author: Professor James Wilson

Version 0:

Reviewer comments:

Reviewer #1

(Remarks to the Author)

Wilson et al presented a protein quantitative trait loci (pQTL) study in the proteomics datasets from Scottish cohort. They discovered 43 possible causal links between protein abundance and disease phenotypes. The experimental design represented a through post GWAS study, however, the conclusions of this study are warranted further validation. Below are my concerns and comments:

1. As the authors mentioned, the subset of 200 samples represents a relatively small size for a pQTL study, and the 505 significant pQTL could be skewed by population sampling bias. The clinical data of all patients should be presented. This is important to understand the statistic power of the analysis. Overall, I think this study is bit overfit for their study models.
2. Given the small sample size, the small number of pQTL hits do need further validation to demonstrate that the identified associations are indeed casual relationships between the protein and risk variants. May be worth to do this in another independent cohort.
3. In general, this study is very descriptive and do not show much scientific impact due to the small sample size for any GWAS study.
4. Beyond increasing sample size, authors should elaborate the proteins with cis and trans pQTL association which should be cross validated.
5. The Somascan has been reported with off-target effect, please discuss this.

Reviewer #2

(Remarks to the Author)

Summary:

Kuliesius and colleagues performed a plasma pQTL study on 6432 proteins measured by SomaScan v4.1 or 7k platform from a Scottish cohort of 200 participants of the European ancestry. Overall, the authors identified 382 cis and 123 trans pQTLs. To connect the protein to diseases, the authors used only the 31 novel cis-pQTLs to perform the Mendelian randomisation and colocalisation. From Mendelian randomization analyses, 149 significant protein-disease pairs were identified. From the follow-up colocalisation analyses, 43 of 149 pairs were also highly colocalized. The authors highlighted a few examples, such as B3GAT1-prostate cancer and LTK-T2D. In general, this study is interesting and will be useful to the pQTL and GWAS field, but I have some comments that should be addressed for this manuscript to be eligible for publication in Communications Biology.

Comments:

1. In the Abstract, line-18, the authors wrote "We discovered 42 colocalising associations", but later the authors reported 43 associations (lines-49, 180, 183). Is this a typo?
2. Definition of novel pQTLs, lines 450-456. In the methods section, the authors only used the proteins targeted by SomaScan v4.0 or 5k platform (Ferkingstad et al 2021 cited as #12) and Olink Explore 1.5k platform (Sun et al 2022 preprint as cited as #15). This strategy has flaws: A) As Sun et al from UKB-PPP consortium published the peer reviewed version with Olink 3k (Explore 3072 PEA) platform, and the authors even cited that version as #6. Why did not consider this newer

- and larger version to define novelty? If you are not going to change your current strategy, please add this as a limitation. B) As the authors cited #8 Surapaneni et al 2022, who used the SomaScan v4.1 or 7k for 466 participants of African ancestry. Could you check how many proteins you listed novel were reported by Surapaneni et al 2022 as significant pQTLs already?
3. The GWAS strategy. The authors seemed to use a genetic relationship matrix as a random effect in their regression model. But as for the 200 participants chosen for this study, the authors reported that the highest genomic relatedness in this sub-cohort was 6% (lines 346-348). Could the authors then clarify the need/necessity to adjusting the genetic relatedness in their model?
 4. The exact numbers of common variants should be listed explicitly, as currently the authors only listed the imputed variants were over 10.5 million (line-56) as the input of the GWAS analyses.
 5. Adding a new main figure at the beginning to list all the analyses in your study design would be beneficial to the readers. Especially you only limited your MR and colocalisation analyses ("Protein – disease links") to the novel cis-pQTLs but not the known+novel cis+trans pQTLs you identified from the "Discovery of pQTLs" section.
 6. There is no method describing how the protein-level variances explained by the different genetic variants were derived. Please add this to the relevant Methods section.
 7. Mendelian randomisation:
 - a. Methods on lines-465 to 466 only mentioned ABO and HLA regions as pleiotropy regions to consider when removing variants as IV, but why not consider the other findings (CFH, HGR, BCHE, VTN, APOE) in this study as well?
 - b. For the pQTL selected as IV, could you provide the F-statistics for each IV used in forward MR? It can further prove if this study provided strong instruments.
 - c. For the outcome disease-set selection. Could you provide the rationale why did you repeat five times with ChatGPT4 to decide if the categorization works? In total, how many outcome traits did you use for the forward MR analyses? The number needs to be added to the results and methods sections of the main text.
 - d. For choosing the significant MR threshold, why did you consider FDR but not Bonferroni correction? Also, why did you consider 0.01 but not 0.05?
 8. Colocalisation: As the authors used coloc R package v5.1.0, the coloc.susie should also be used along with the current coloc.abf results. The coloc.susie method could capture more than one causal variant shared by the protein and disease in theory, as coloc.abf limits such possibilities.
 9. If possible, I would suggest the authors performing a power analysis based on the effect size of pQTLs and sample size of this study, rather than using the analyses reported by Sun et al 2023 from the UKB-PPP study (line-228). Sun et al 2023 found sample size ≥ 5000 would be necessary for cis pQTLs to plateau, while the 200 is much fewer than this estimation.
 10. GO enrichment for all genes/proteins associated with the six trans pleiotropic regions/hubs. Please use only 6432 proteins measured by the SomaScan v4.1 or 7k platform as the background reference list. The usage of the entire 20k human genes (line-421) as background is not the correct choice in this scenario.
 11. Data availability. I cannot find this study on the webpage, please make sure it is indeed uploaded.

Version 1:

Reviewer comments:

Reviewer #1

(Remarks to the Author)

I appreciate the thoughtful responses and editing. The authors have fully addressed my concerns and comments. A proofread is needed to eliminate any typo as Reviewer 2 suggested.

Reviewer #2

(Remarks to the Author)

Summary:

The authors have addressed comments by the reviewers, especially for the new Figure 1 summarizing the study design improved the manuscript. Below are my additional comments:

Comments:

1. The author defined the protein-level variances explained by the different genetic variants that were derived with the formula from line 534-539,
 - a. While the term β^2 makes sense, could the authors clarify why they considered the terms MAF as p and $(1-MAF)$ as q in this variance estimation?
 - b. Could you please explain why you chose to use the summary-level data to derive the variance instead of using the individual-level data (for example, the adjusted R-square from the regression modeling the protein-variant association)?
2. For Mendelian randomization, the main text mentions 3772 outcome traits, but Supplementary Table 11 lists 4904 outcome traits and Supplementary Table 12 lists 4604 outcome traits. Could the authors explain these discrepancies?
3. The authors cited Katz et al 2020 (<https://www.science.org/doi/10.1126/sciadv.abm5164>) for addressing the power analysis on pQTL identification from the previous comment 9. However, I would like to point out that the original Figure 2B from Katz et al illustrates the reproducibility of sample-level protein differences between groups, rather than directly addressing variant-protein associations (or pQTL estimates) in this study. Protein level differences can be influenced by multiple genetic and non-genetic factors. Additionally, while Katz et al discusses pQTL comparisons between platforms, they do not specifically perform power analysis for pQTL analysis.

4. The order of the five supplementary figures needs to be corrected; they are currently arranged as 5/3/2/4/1.
5. Considering that the title of the manuscript includes "drug target", I have additional comments for the authors to investigate. While the authors have mapped proteins to drug compounds from the DrugBank, they have not included any information on indications or side effects. This information is crucial as it serves as a source of indirect validation and offers further evidence for potential repurposing. Could this be added? For example, ChEMBL contains relevant information on this.

Version 2:

Reviewer comments:

Reviewer #2

(Remarks to the Author)

The authors have fully addressed my comments.

Reviewer 1

1. As the authors mentioned, the subset of 200 samples represents a relatively small size for a pQTL study, and the 505 significant pQTL could be skewed by population sampling bias. The clinical data of all patients should be presented. This is important to understand the statistic power of the analysis. Overall, I think this study is bit overfit for their study models.

We thank the reviewer for the insightful feedback. We agree that smaller cohort studies, like the one in this study, can indeed introduce unique challenges, while at the same time providing valuable opportunities. For example, small isolated cohort studies could prove valuable in identifying the impact of some rare variants. Random genetic drift leads to their overrepresentation in such populations, which increases the power to detect association.

For instance, as the reviewer inferred, while our cohort used population-based sampling of generally healthy individuals, it provides excellent context for studying disease-relevant genetic variations in the general population. One recent example of this is the finding that the oncogenic BRCA1 variant is encountered with 480 times the frequency in the Orkney Isles when compared to the broader UK population (<https://doi.org/10.1038/s41431-023-01297-w>), demonstrating the potential for significant findings even in smaller, isolated cohorts.

Unfortunately, there is neither Research Ethics Committee approval, nor consent from individual participants, to permit open release of the individual-level research data underlying this study (see Data Availability section of the manuscript). As a compromise, we have now included Supplementary Table 10, which provides a summary of key medical information for the 200 participants in this study, including age, sex, BMI, and other health-related data split by sex.

Finally, we have taken every precaution to prevent overfitting, from rigorous pre-GWAS quality control to strict false positive filtering in post-GWAS analyses, and now independent replication of our findings. We believe our approach has yielded robust statistical results and offers valuable insights for identifying new drug targets and have added additional commentary on this in the Discussion section of the manuscript (lines 361-376).

2. Given the small sample size, the small number of pQTL hits do need further validation to demonstrate that the identified associations are indeed casual relationships between the protein and risk variants. May be worth to do this in another independent cohort.

We thank the reviewer for raising this important point regarding the need for further validation. Since we do not have access to another independent cohort with a similar data, or funds to measure proteomics in more samples from our own cohort, we put in a considerable effort to replicate our findings in publicly available cohorts. First, we assessed the replication of pQTL for 272 proteins that were measured in an

independent cohort, with an earlier version of the SomaLogic assay (<https://doi.org/10.1126/science.abj1541>). Not only were the *cis* pQTL we report here replicated to a high degree, with a strong correlation in effect size ($r^2 = 0.96$) between the two studies, but the effect sizes were also fully consistent in directionality. Further details on this replication are provided in the Methods section (lines 557-581), and the full results can be found in Supplementary Table 1 and Supplementary figure 5.

In addition, we conducted a separate validation procedure for the 14 protein targets that we annotated with causal associations. For this, we used the only other available SomaLogic v4.1 dataset, along with the recently published full Olink Explore 3072 dataset in UK Biobank and eQTL data from eQTLGen, to assess whether the signals in these datasets colocalize with those reported in our study (Supplementary Table 8).

Finally, we replicated our MR findings with external data, either by using independent instruments (pQTL from another SomaLogic dataset, Olink dataset, or eQTL from eQTLGen), or by using independent outcome data, or both, where possible (Supplementary Tables 7 and 13). A summary of our replication results is included in Supplementary Table 5.

Overall, we successfully replicated 31 of the 43 initially reported protein-disease links. Of the 12 that were not replicated, 10 were related to blood cell counts. We have discussed these findings, along with replication in general, in the Discussion section (lines 366-376), as well as in the results sections related to B3GAT1 and NTAQ1 (lines 232-236 & 254-256).

3. In general, this study is very descriptive and do not show much scientific impact due to the small sample size for any GWAS study.

We thank the reviewer for their thoughtful feedback. We understand the concern regarding the sample size and its implications for GWAS studies. In the era of ever-increasing GWAS sample sizes and their associated costs, we have advocated for the utility of smaller cohort studies, especially in the context of proteomics. Proteins, as molecular markers, are closer to the genotype in the biological pathway, thereby often showing stronger and more direct genetic associations. This proximity reduces the complexity and the number of confounding variables, enhancing the statistical power, even with smaller sample sizes.

To further support this point, we have added a new reference (Katz et al. 2022, Figure 2B), which demonstrates that a sample size of 200 can reliably detect approximately a 5% change in protein levels using an earlier version of the SomaLogic assay. Our findings align with this, as we observe a minimum protein level change of 7% associated with a sentinel pQTL among our results (see Supplementary Table 1, Variance Explained column). We have referenced this in the main text (lines 280-286).

By using robust methodology, we have not only highlighted three new potential opportunities for drug development for prostate cancer, type-2 diabetes, and macular

degeneration, but also identified four other causal genetic links to outcomes of medical interest. These findings underscore the effectiveness of focused proteomic studies, which can yield significant insights and tangible impacts in medical research with potentially lower numbers of subjects.

This approach is particularly useful in a time when the scalability of large studies may be limited by resources or at the advent of new technologies. Furthermore, our replication results, detailed in Supplementary Tables 1 and 5, illustrate that even with a modest sample size, robust and significant scientific contributions can be made.

4. Beyond increasing sample size, authors should elaborate the proteins with *cis* and *trans* pQTL association which should be cross validated.

We thank the reviewer for suggesting replication of our pQTL associations. To validate our *cis* pQTL findings, we compared them to GWAS summary statistics from Pietzner et al. (2021), which measured proteins using the earlier SomaScan v4.0 assay in over 10,000 individuals. Of the 272 proteins with *cis* pQTL in our study that overlapped with theirs, all were successfully replicated, meeting genome-wide significance and showing consistent effect size directionality. Furthermore, we cross-validated our results using the only other SomaLogic v4.1 study by Surapaneni et al. (2022), assessing linkage disequilibrium (LD) between sentinel pQTL using a UK Biobank reference panel (Supplementary Table 2). These replication efforts are detailed in lines 73-91 & 351-360.

Then, we assessed the colocalisation between the genetic signals identified through other published SomaScan, Olink and RNA sequencing assays (Supplementary Table 13, lines 653-659). We have not been able to colocalise 5 of the 14 proteins with any of these assays, likely owing to differences in genetic ancestry, the lack of overlap between pQTL and eQTL and the general differences between proteomic assays.

5. The Somascan has been reported with off-target effect, please discuss this.

We thank the reviewer for raising this important point regarding the potential off-target effects of the SomaScan assay. We agree that this is a key consideration, and both our own findings—particularly in the results section discussing FCGR2A, FCGR2B, and FCGR2C—and those reported by others have highlighted some discrepancies between the genomic locations of genes encoding specific proteins and the associations identified via GWAS. Typically, one might expect most protein expression levels to associate with genetic variation within their exonic, intronic, or nearby regulatory regions. However, research indicates that less than half of the assayed proteins in the SomaLogic assay have a *cis* pQTL (<https://doi.org/10.1038/s41586-023-06563-x>). This discrepancy suggests potential off-target effects of the assay.

In light of these observations, we have chosen to focus primarily on *cis* effects in this study as they are less likely to be incidental and can provide more reliable evidence that the assay is measuring the intended proteins – it is, after all, much less likely

that a SomaScan assay designed to measure protein 1, but in fact only targeting protein 2, is associated with genetic variation in the gene encoding protein 1 (compared to the scenario where the assay *is* measuring the protein SomaLogic claims it does and which is encoded by the gene in which a genome-wide significant pQTL is observed). We believe that focusing on *cis* pQTL helps mitigate the impact of any off-target effects of the SomaScan assay. We have expanded on this point in the results section of the manuscript, where we explain our strategy for addressing pleiotropy by exclusively using *cis* instruments in our post-GWAS analyses (lines 176-180 & 403-406).

Reviewer 2

1. In the Abstract, line-18, the authors wrote “We discovered 42 colocalising associations”, but later the authors reported 43 associations (lines-49, 180, 183). Is this a typo?

We thank the reviewer for pointing this out. Indeed, there is a typo in the abstract. Referencing Supplementary Table 6, one can find 43 colocalising associations with $PP.H4.abf > 0.8$. We have correctly referenced the 43 associations throughout the rest of the text, and we have updated the abstract to reflect the correct number. We sincerely appreciate your careful attention to detail.

2. Definition of novel pQTLs, lines 450-456. In the methods section, the authors only used the proteins targeted by SomaScan v4.0 or 5k platform (Ferkingstad et al 2021 cited as #12) and Olink Explore 1.5k platform (Sun et al 2022 preprint as cited as #15). This strategy has flaws: A) As Sun et al from UKB-PPP consortium published the peer reviewed version with Olink 3k (Explore 3072 PEA) platform, and the authors even cited that version as #6. Why did not consider this newer and larger version to define novelty? If you are not going to change your current strategy, please add this as a limitation. B) As the authors cited #8 Surapaneni et al 2022, who used the SomaScan v4.1 or 7k for 466 participants of African ancestry. Could you check how many proteins you listed novel were reported by Surapaneni et al 2022 as significant pQTLs already?

We thank the reviewer for their detailed feedback. We appreciate the opportunity to clarify these points.

2A. Active research took place in-between the publication of the Sun et al 2022 preprint and Sun et al 2023 final version of the UKB-PPP consortium paper, which expanded on their initially reported protein panel. To the best of our knowledge, there were no publications with data from the Olink Explore 3072 platform at the time of our Mendelian Randomization instrument selection (April 2023). Since our initial submission to Communications Biology, the proteomics assays offered by SomaLogic and Olink have since grown further in the number of proteins measured. We acknowledge that defining the unstudied proteome is increasingly complex in this rapidly evolving field. We have now added this as a limitation of our study and appreciate your understanding (lines 396-406).

2B. Of the 382 *cis* pQTL reported in this study, 156 (41%) overlap with *cis* pQTL in the Surapaneni et al 2022 study, defined as reporting a genome-wide significant pQTL within LD $r^2 > 0.6$ of the sentinel SNP in VIKING, using the reference panel of a random subset of 10,000 unrelated genomically British individuals from UK Biobank (Supplementary table 2). Of the 31 proteins with *cis* pQTL we selected for Mendelian Randomization, each with a single sentinel variant, 13 (42%) also met the same criterion. This is now described in lines 351-360.

We do not think that this takes away from this subset of proteins to be used as new instruments, given the discrepancy between reported genome-wide significant

variants in the two studies. The differences in ancestry between the African American cohort used in their study and the isolated Scottish cohort used in ours may result in variations in pQTL signals. For instance, the genetic signal for PROCR protein measurements does not colocalise between the two studies, with $H3 > 99\%$ (Supplementary Table 13).

3. The GWAS strategy. The authors seemed to use a genetic relationship matrix as a random effect in their regression model. But as for the 200 participants chosen for this study, the authors reported that the highest genomic relatedness in this sub-cohort was 6% (lines 346-348). Could the authors then clarify the need/necessity to adjusting the genetic relatedness in their model?

We thank the reviewer for their thoughtful question. We chose to include the genetic relationship matrix as a random effect in the regression model for several reasons:

1. Correction for population stratification – Subtle population structure arising from large family kindreds may still exist in cohorts with relatedness thresholded at 6%. This is particularly important given our low GWAS sample size, where even small amounts of such relatedness could impact the results.
2. Standard practice – The use of a genetic relationship matrix to correct for inflation due to familial structure is standard practice in GWAS. While we did not specifically investigate the potential consequences of omitting this adjustment in a cohort thresholded at 6% relatedness, we chose to follow this more conservative approach to minimize the risk of false-positive associations.

We would also like to note that a Scree plot of the principal components of the genetic relationship matrix (Supplementary figure 1) identifies a clear inflection point, which allowed us to use just three genetic principal components instead of the commonly seen 10 or more in GWAS literature.

4. The exact numbers of common variants should be listed explicitly, as currently the authors only listed the imputed variants were over 10.5 million (line-56) as the input of the GWAS analyses.

We thank the reviewer for their helpful suggestion. We agree that explicitly listing the total number of common variants would provide greater clarity, especially since we purposefully exclude rare variants from the analysis, due to low sample size. There are, in fact, a total of 5,462,684 variants with minor allele frequency of ≥ 0.05 in the VIKING 200 sub cohort with proteomics data. We have now added this information to the methods section for transparency and completeness (lines 495-500).

5. Adding a new main figure at the beginning to list all the analyses in your study design would be beneficial to the readers. Especially you only limited your MR and colocalisation analyses (“Protein – disease links”) to the novel

cis-pQTLs but not the known+novel cis+trans pQTLs you identified from the “Discovery of pQTLs” section.

We thank the reviewer for this suggestion. We agree that a study design figure will be highly beneficial for readers. We have now added a such a figure (Figure 1), which clearly delineates the three post-GWAS result analysis groups and gives a high-level overview of the research plan.

6. There is no method describing how the protein-level variances explained by the different genetic variants were derived. Please add this to the relevant Methods section.

We thank the reviewer for bringing this to our attention. We have now added a detailed explanation in the "Independent associations and LD proxies" section of the Methods, outlining how the variance in protein levels explained by different genetic variants was calculated using the SNP allele frequency and effect size estimates (lines 534-539).

7. Mendelian randomisation:

a. Methods on lines-465 to 466 only mentioned ABO and HLA regions as pleiotropy regions to consider when removing variants as IV, but why not consider the other findings (CFH, HGR, BCHE, VTN, APOE) in this study as well?

b. For the pQTL selected as IV, could you provide the F-statistics for each IV used in forward MR? It can further prove if this study provided strong instruments.

c. For the outcome disease-set selection. Could you provide the rationale why did you repeat five times with ChatGPT4 to decide if the categorization works? In total, how many outcome traits did you use for the forward MR analyses? The number needs to be added to the results and methods sections of the main text.

d. For choosing the significant MR threshold, why did you consider FDR but not Bonferroni correction? Also, why did you consider 0.01 but not 0.05?

7A. We appreciate the reviewer’s observation and would like to clarify the distinction between the extensive genetic polymorphisms in the ABO and HLA regions and the pleiotropic effects observed in the five genetic regions reported in this study. The ABO and HLA regions are particularly challenging to analyse due to extended regions of high linkage disequilibrium (LD), which can complicate interpretation. In particular, it can be difficult to annotate an association as cis or trans, since many SNPs are in high LD in large windows. In contrast, the five regions identified in our study that exhibit pleiotropic effects on protein levels are in regions of lower polymorphism and LD. We have now clarified this further in the main text (lines 589-592) and added a supporting citation (citation number 75). Additionally, upon further investigation, none of the 31 selected MR instruments were found within 1 Mb of the transcription start of the five identified trans hub genes, confirming that they are unlikely to be influenced by these hubs.

7B. We thank the reviewer for the suggestion. We have now added the instrumental variable F-statistic column in the Supplementary 1 and 6 tables. The weakest pQTL instrumental variable used in forward MR had an F-statistic of 31.9, indicating that all

instruments used in this study meet the criteria for strong instruments (lines 593-597).

7C. The use of Large Language Models (LLMs) in research was relatively novel at the time we conducted this study, occurring just four months after the release of ChatGPT4. Given the generative nature of these models, they can be highly versatile but may yield slightly different results when categorizing items, especially when items could reasonably belong to multiple categories. To address this, we opted to run the model five times and use a consensus approach to improve the accuracy of our outcome trait categorization.

The final OpenGWAS outcome trait categorization was included in the Supplementary Table 12, and we have now also stated the number explicitly in the appropriate results and methods sections (lines 606-608).

7D. We chose to use the False Discovery Rate (FDR) correction over Bonferroni correction due to the nature of both our instruments and outcome data. Bonferroni correction, while effective in controlling false positives, is often too stringent when the tests are not independent, leading to a higher rate of false negatives. Both our instrumental variable data (proteins) and outcome data (traits/diseases) are not all independent. For example, leg fat percentage and whole-body fat percentage would be considered as independent tests if we utilised the Bonferroni correction, while these two traits are clearly related. We, therefore, decided on a compromise – using a less conservative FDR correction, but somewhat more conservative alpha significance threshold. We have also now clarified this reasoning in the Methods (lines 609-615).

8. Colocalisation: As the authors used coloc R package v5.1.0, the coloc.susie should also be used along with the current coloc.abf results. The coloc.susie method could capture more than one causal variant shared by the protein and disease in theory, as coloc.abf limits such possibilities.

We thank the reviewer for the insightful comment. While we recognize the advantages of the SuSiE method in capturing multiple causal variants in regions with complex genetic architectures, the protein levels in our study are each influenced by a single sentinel pQTL, as detailed in Supplementary Table 1. In this context, the coloc.abf method is particularly well-suited, providing efficient and robust results when a single causal variant is driving the association.

Nonetheless, we acknowledge that the outcome dataset (disease or medical trait GWAS) may contain multiple causal variants, where coloc.susie could be more appropriate. However, due to previous challenges in running SuSiE, we opted for coloc.abf as a conservative approach. We appreciate that this choice may result in some signals not colocalizing due to methodological limitations and we clarify this in the main text (lines 204-208).

9. If possible, I would suggest the authors performing a power analysis based on the effect size of pQTLs and sample size of this study, rather than using the analyses reported by Sun et al 2023 from the UKB-PPP study (line-228). Sun et al 2023 found sample size ≥ 5000 would be necessary for cis pQTLs to plateau, while the 200 is much fewer than this estimation.

We apologize for any confusion in our previous wording. We did not intend to imply that the *cis* pQTL identified in our study had reached a plateau. The reference to Sun et al. 2023 was meant to highlight the differences in the proportion of *cis* pQTL reported in our study compared to other large-scale proteomics studies. To avoid further ambiguity, we have now revised the text to clarify this point (lines 386-395).

We have also added a new reference which includes a power analysis (Figure 2B in Katz et al. 2022) demonstrating that a sample size of 200 can detect approximately a 5% change in protein levels using the older version of the SomaLogic assay. Our data supports this finding, as we observe a minimum protein level change of 7% associated with a sentinel pQTL (see Supplementary Table 1, Variance Explained column). Additionally, we have been informed that the updated SomaLogic assay has maintained its low variability while expanding the number of targets (<http://dx.doi.org/10.1021/acs.jproteome.4c00667>).

10. GO enrichment for all genes/proteins associated with the six trans pleiotropic regions/hubs. Please use only 6432 proteins measured by the SomaScan v4.1 or 7k platform as the background reference list. The usage of the entire 20k human genes (line-421) as background is not the correct choice in this scenario.

We thank the reviewer for this valuable comment. We have re-performed the GO enrichment analysis using the 6,432 proteins measured by the SomaScan v4.1 platform as the background reference list, as recommended. Encouragingly, the enrichment patterns remained consistent with our initial findings. The methods section (lines 510-516) and the associated Supplementary Table 9 have been updated accordingly to reflect this updated analysis.

11. Data availability. I cannot find this study on the webpage, please make sure it is indeed uploaded.

We thank the reviewer for bringing this to our attention. We were in the process of uploading the data but encountered some challenges with hosting the multi-terabyte dataset on our local sharing service. To resolve this, we have now made the data available on the GWAS Catalog, which is specifically suited for this purpose. The study can be accessed via the following Accession IDs: GCST90436912 – GCST90444200. We have also updated the data availability section of the manuscript to reflect this.

1. The author defined the protein-level variances explained by the different genetic variants that were derived with the formula from line 534-539,

a. While the term β^2 makes sense, could the authors clarify why they considered the terms MAF as p and $(1-MAF)$ as q in this variance estimation?

b. Could you please explain why you chose to use the summary-level data to derive the variance instead of using the individual-level data (for example, the adjusted R-square from the regression modeling the protein-variant association)?

1a. We sincerely thank the reviewer for their careful attention to detail in identifying the need for clarification regarding variable definitions in our variance explained formula. We have corrected the variable descriptions in the text (lines 539-544) to ensure consistency with the standard genetic notation. This change does not alter the results, as the variance explained formula accounts for allele frequencies symmetrically.

1b. We agree with the reviewer that individual-level regression model (e.g., adjusted R^2) could provide an alternative, more accurate estimate of variance explained in this study. However, the formula we employed ($\beta^2 * 2 * p * q$) allows for direct comparison with other published studies (lines 386-390; reference 52 in the manuscript), where we don't have access to individual level data. We therefore chose to use this approximation rather than more precise estimates calculated on the individual level data.

2. For Mendelian randomization, the main text mentions 3772 outcome traits, but Supplementary Table 11 lists 4904 outcome traits and Supplementary Table 12 lists 4604 outcome traits. Could the authors explain these discrepancies?

To clarify our methodology: from the 4,904 GWAS datasets available in the GWAS Catalog at the time of analysis, we manually curated and retained 4,604 datasets that met our quality criteria. Of these, 3,772 datasets were subsequently included in our Mendelian randomization analyses. We have now provided an explanation with greater details in the MR methodology section (lines 605; 613-625) to improve transparency. We are grateful to the reviewer for highlighting this important methodological point, which has strengthened the clarity of our manuscript.

3. The authors cited Katz et al 2020

(<https://www.science.org/doi/10.1126/sciadv.abm5164>) for addressing the power analysis on pQTL identification from the previous comment 9. However, I would like to point out that the original Figure 2B from Katz et al illustrates the reproducibility of sample-level protein differences between groups, rather than directly addressing variant-protein associations (or pQTL estimates) in this study. Protein level differences can be influenced by multiple genetic and non-genetic factors. Additionally, while Katz et al discusses pQTL comparisons between platforms, they do not specifically perform power analysis for pQTL analysis.

We thank the reviewer for highlighting this important distinction. While our previous reference demonstrated technical reproducibility of protein measurements, we agree that their analysis did not explicitly address statistical power for pQTL identification, and it has now been referenced appropriately (lines 382-383). To this end, we have now performed a dedicated power analysis for pQTL detection in our study.

The new Supplementary Figure 4 illustrates these relationships in lines 280-283 and adding further context in lines 383-386. We hope this clarifies our study's capacity to detect robust genetic effects.

4. The order of the five supplementary figures needs to be corrected; they are currently arranged as 5/3/2/4/1.

Thank you to the reviewer for highlighting this error in the supplementary figure ordering. We have now corrected the sequence of all supplementary figures throughout the manuscript text and in the supplementary materials files to reflect the order of appearance in the text (1-6). The figures and their corresponding in-text citations have been carefully checked to ensure consistency.

5. Considering that the title of the manuscript includes "drug target", I have additional comments for the authors to investigate. While the authors have mapped proteins to drug compounds from the DrugBank, they have not included any information on indications or side effects. This information is crucial as it serves as a source of indirect validation and offers further evidence for potential repurposing. Could this be added? For example, ChEMBL contains relevant information on this.

We appreciate the reviewer's suggestion to strengthen the drug target validation in our study by incorporating therapeutic indications and safety profiles. In response, we have now added a new supplementary table 10 with relevant annotations from DrugBank and ChEMBL for the 14 proteins with medically relevant trait associations in this study. We also highlighted these findings in the text (lines 348-353).